# Ultra-compact dual-band smart NEMS magnetoelectric antennas for simultaneous wireless energy harvesting and magnetic field sensing

Mohsen Zaeimbashi [1], Mehdi Nasrollahpour[1], Adam Khalifa[2], Anthony Romano[1], Xianfeng Liang[1], Huaihao Chen[1], Neville Sun[1], Alexei Matyushov [1,3], Hwaider Lin[1], Cunzheng Dong [1], Ziyue Xu[1], Ankit Mittal [1], Isabel Martos-Repath[1], Gaurav Jha[1], Nikita Mirchandani[1], Diptashree Das[1], Marvin Onabajo[1], Aatmesh Shrivastava[1], Sydney Cash[2] & Nian X. Sun [1,4,5 ✉]

Ultra-compact wireless implantable medical devices are in great demand for healthcare applications, in particular for neural recording and stimulation. Current implantable technologies based on miniaturized micro-coils suffer from low wireless power transfer efficiency (PTE) and are not always compliant with the specific absorption rate imposed by the Federal Communications Commission. Moreover, current implantable devices are reliant on differential recording of voltage or current across space and require direct contact between electrode and tissue. Here, we show an ultra-compact dual-band smart nanoelectromechanical systems magnetoelectric (ME) antenna with a size of $250 \times 174\ \mu m^2$ that can efficiently perform wireless energy harvesting and sense ultra-small magnetic fields. The proposed ME antenna has a wireless PTE 1–2 orders of magnitude higher than any other reported miniaturized micro-coil, allowing the wireless IMDs to be compliant with the SAR limit. Furthermore, the antenna's magnetic field detectivity of 300–500 pT allows the IMDs to record neural magnetic fields.

[1] Department of Electrical and Computer Engineering, Northeastern University, Boston, MA, USA. [2] Department of Neurology, Massachusetts General Hospital, Harvard Medical School, Boston, MA, USA. [3] Department of Physics, Northeastern University, Boston, MA, USA. [4] Department of Chemical Engineering, Northeastern University, Boston, MA, USA. [5] Department of Bioengineering, Northeastern University, Boston, MA, USA. ✉email: nian@ece.neu.edu

Direct measurement of widespread neuronal activity at the level of small ensembles of neurons is essential for both basic research and clinical applications. Current technology for doing so exists, but is limited in the number of sites and spatial extent coverable. To address the need for a distributed neural interfacing tool, a new generation of wirelessly powered standalone implantable medical devices (IMDs) have emerged in the last few years[1–4]. Wirelessly powered IMDs eliminate the invasiveness and discomfort caused by batteries and wires in most conventional implants. The two major wireless powering modalities today are electromagnetic and ultrasonic. Regardless of the powering method used, as the receiver shrinks, it becomes increasingly difficult to deliver sufficient power to operate the IMD, which can range from a few to several hundred microwatts depending on the application. This puts a heavy burden on the miniaturization of coils and piezo devices, which explains why many state-of-the-art single channel devices remain bulky[5–9]. Ultimately, these devices will need to be surgically implanted into the brain but the technology remains too large for human applications. Some untethered IMDs have successfully miniaturized their transducers[10–14] down to the scale of a human hair diameter. However, the current prototypes do not have the necessary wireless efficiency to be safely powered in human applications because the SAR limit imposed by the FCC needs to be respected. In this article, an acoustically actuated nano-mechanical ME antenna with the highest power transfer efficiency reported to date is described. These ME antennas incorporate a magnetic and piezoelectric heterostructure where the magnetic film senses H-components of EM waves. The magnetic layer then induces an oscillating strain, which generates a piezoelectric voltage output at the electromechanical resonance frequency. By exploiting this transduction mechanism, ME antennas do not suffer from the same miniaturization constraints as coils, and they can be driven by weak magnetic fields[15].

We have most recently reported magnetoelectric (ME) nanoelectromechanical system (NEMS) resonators as single-band ME antennas for wireless communication[15], and ME sensors for quasi-static low-frequency magnetic field sensing[16] in two separate devices with two different designs and different operating frequencies. The ME antenna is based on ME FBAR (thin film bulk acoustic wave resonator) working at 2.5 GHz; while the ME sensor is based on a ME CMR (contour mode resonator) with interdigitated electrode and an operation frequency of 215 MHz. In this work we present smart ME antenna with unprecedented characteristics that are ideal for IMDs: (1) ultra-compact antenna for highly efficient wireless power transfer efficiency and data communication at GHz; (2) ultra-sensitive magnetometer capable of sensing picoTesla low-frequency fields by using MHz resonance; and (3) simultaneous operation at two different frequency bands, GHz for wireless power transfer and data communication, and MHz for magnetic field sensing. Compared to neural electrical sensing based on differential voltages from neural probes or wireless implants[17], the proposed smart ME antennas operate based on neuronal magnetic sensing, providing multiple advantages: (1) magnetic neural sensing is not differential, which enables more compact neural recording implants or elements[17]; (2) thanks to its physical characteristics, magnetic neural sensing is more localized than electrical sensing, which consequently enables better spatial resolution of neuronal activities; (3) it is practical to create safe and contact-less implants coated with bio-compatible polymer films such as parylene, unlike in conventional devices where the necessary direct contact between electrode and tissue degrades over time due to electrochemical fouling and tissue reactions; (4) the same technology can be used for animals and human, allowing for direct comparisons and easier translation of animal to human information; and (5)

compact size allows for distributed, addressable, high channel count use in the parenchyma, pial surface, extra dural, and in both central and peripheral nervous tissue[18]. In the following sections, we will discuss the design and characteristics of the proposed smart ME antenna, its wireless energy harvesting performance, its magnetic field sensing capability, and the device's performance for simultaneous energy harvesting and magnetic field sensing. At the end, we will discuss future works and challenges and propose a circuit design for autonomous data communication and energy harvesting.

## Results

**Design and characteristics of smart ME antenna.** Magneto-electric antennas are based on multiferroic materials, which are materials that show more than one of the primary ferroics properties. These primary ferroics include: (a) ferromagnetism, a phenomenon in which magnetization can be changed by an applied magnetic field; (b) ferroelectricity, in which electrical polarization can be altered by an applied electric field; (c) fer-roelasticity, in which deformation can be changed by an applied stress. ME antennas, consisting both piezoelectric and magne-tostrictive thin-film materials, exploit the aforementioned ferroic properties in order to receive and transmit electromagnetic waves. Piezoelectric thin-film couples electrical polarization and mechanical strain, and magnetostrictive material couples magnetic polarization and mechanical strain. A piezoelectric and magnetostrictive heterostructure, deposited by sputtering system is used to couple the discussed ferroic orders. The diagram in Fig. 1a visualizes the concepts of the transmitter ($T_X$) and receiver (Rx) operating modes of an ME antenna, where the top and bottom rectangular boxes represent the mechanically-coupled magnetostrctive and piezoelectric layers, respectively. In $T_X$ mode (L), an RF voltage is applied to piezo material, which in turn generates mechanical strain. This mechanical strain is then transferred to the magnetostrictive thin-film, which subsequently radiates EM wave due to a piezomagnetic phenomenon. In Rx mode (R), the same sequence occurs in reversed order, where incoming EM wave towards magnetic material generates strain. The strain is then transferred to the piezoelectric thin-film, which in turn generates an RF voltage in the output. The simulation results of strain and displacement distribution in magnetostrctive and piezoelectric layers are shown in Appendix A. The incoming EM wave produces a strain in magnetostrctive layer; the strain is then transferred to piezoelectric layer, which in turn induces a voltage across its thickness that can be used for energy harvesting purposes.

Figure 1b and c shows the 3D schematic of the ME antenna from this work, for which fabrication steps are summarized in Appendix B. This ME device consists of three parallel rectangular ME resonators, each with a size of $250 \times 50\ \mu m^2$, giving an overall size of $250 \times 174\ \mu m^2$, including two 12 μm gaps between resonators. The thickness of the resonators, excluding the 50 nm Pt layer, is 1 μm which consists of AlN and FeGaB layers each 500 nm in thickness. We have designed ME antennas with one to seven number of parallel resonators. Our experiments have shown that the energy harvesting and magnetic field sensing performance of the ME antenna enhance by increasing the number of parallel resonators. This is due to the increased effective area of the antenna that is in presence of external magnetic field. Even though using an antenna with seven parallel resonators will have a better performance, the size of antenna will consequently increase, too—which is not desirable for our target application. Thus, in this project we have selected an antenna with three parallel resonators to have a relatively small size and good energy harvesting and magnetic field sensing performance.

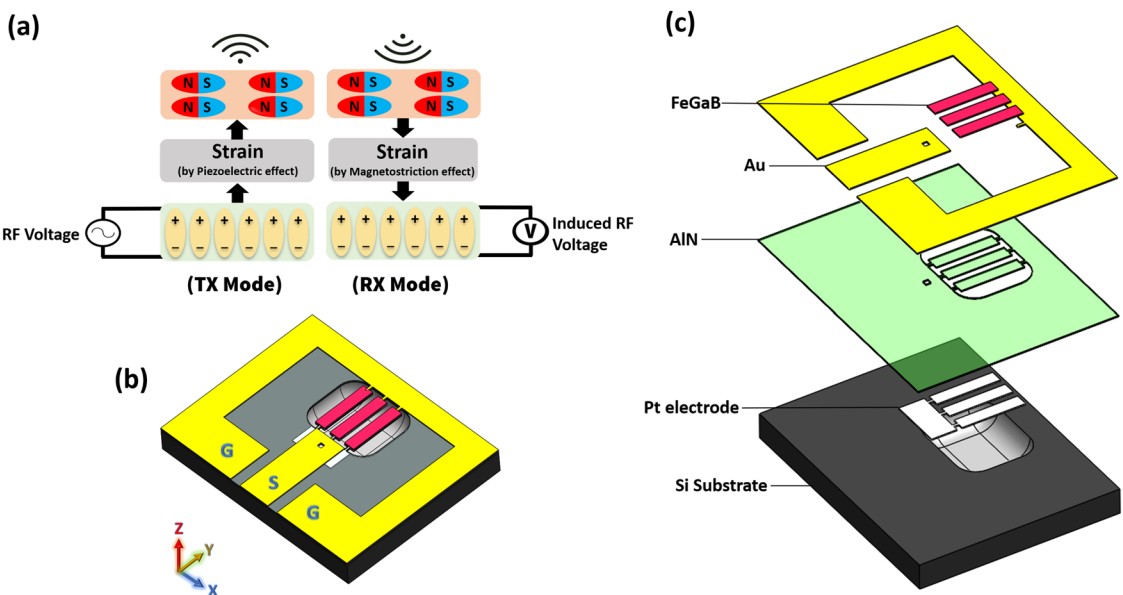

**Fig. 1 3D structure and operating concept of the proposed ME antenna. a** Diagram showing the concept of Tx and Rx operating modes of ME antenna, where the top and bottom rectangular boxes represent the magnetostrctive and piezoelectric layers, respectively; **b** 3D schematic of the ME antenna on released Si substrate. The yellow color indicates a gold ground ring and GSG pads that are used for probing and wire-bonding the antenna to PCB; **c** exploded structure of the ME antenna showing all the layers.

The Si substrate underneath the ME resonators is etched in order to release the ME elements and prevent damping of the acoustic vibration, a process that improves the performance of ME antenna. The resonance frequency of the rectangular ME elements is defined by the size of piezoelectric thin film materials. AlN was used as the piezoelectric phase and FeGaB as magnetostrictive phase that are mechanically coupled through the interface. A rectangular thin-film piezoelectric material can exhibit three resonance frequencies: along width, length, and thickness. Here, we use the width and thickness modes of the ME resonators since they are more efficient than the length mode. Therefore, the proposed ME antenna has two operational resonance frequencies: (1) a contour mode resonator (CMR) frequency corresponding to the width mode of vibration in piezoelectric thin-film, and can be expressed as $f_{r,CMR} \sim \frac{1}{2W_0} \sqrt{\frac{E_{eq}}{\rho_{eq}}}$, where $W_0$ is the width of the resonator, and $E_{eq}$ and $\rho_{eq}$ are the equivalent Young's modulus and equivalent density of the resonator stack, respectively; (2) a thin-film bulk acoustic wave resonator (FBAR) frequency associated with the thickness mode of vibration applied to the resonator, which can be expressed as $f_{r,FBAR} \sim \frac{1}{2t} \sqrt{\frac{E_{eq}}{\rho_{eq}}}$, where t is the thickness of the AlN thin-film. Since the designed antenna operates in two different frequency bands, we refer to this device a smart or hybrid FBAR/CMR ME antenna. It is notable that since the thickness of the AlN film is much smaller than its width, the thickness mode (FBAR) resonance frequency of the ME antenna is significantly higher than the width mode (CMR) resonance frequency.

Figure 2a shows an optical microscope image of a fabricated smart ME antenna with three parallel ME resonators connected to a gold ground ring with GSG probing pads. Three parallel ME resonators were used for enhanced wireless energy transfer efficiency without significantly enhanced device size. The red color is due to the reflection from the AlN thin-film because the Si substrate underneath is etched. Figure 2b displays the measured $S_{11}$ of the thickness mode of the smart ME antenna showing a resonance frequency at 2.51 GHz, which corresponds

to 500 nm thickness of AlN. The two dips are slightly apart from each other as a result of the small difference in the stress levels of the three rectangular ME resonator array. Figure 2c shows the $S_{11}$ of the width mode showing a resonance frequency at 63.6 MHz, which corresponds to 50 µm width of AlN. It is noteworthy that the $S_{11}$ plots of both thickness and width modes were measured after wire bonding the ME antenna to a 200 µm thick paper PCB, on which an SMA connector is mounted in order to connect the ME antenna to a vector network analyzer (VNA). We observed some impedance matching ($S_{11}$) degradation after the wire bonding process compared to the case of directly probing ME antenna pads using standard GSG probes. This change is attributed to the extra series inductance and resistance added by bonding wires. The insets in Fig. 2b and c shows the equivalent Modified Butterworth–Van Dyke (MBVD) circuit model for thickness and width modes of ME antenna, respectively. In the following sections it will be shown that thickness and width modes of smart ME antenna exhibit a high performance while harvesting the RF energy and sensing magnetic fields, respectively —functionalities that can be utilized in ultra-miniaturized brain and body implantable devices and wearable technologies.

**Wireless energy harvesting with smart ME antenna.** An ME antenna's energy harvesting mode functions based on the Rx mode diagram in Fig. 1a, where an incoming radio frequency magnetic field induces strain in the magnetostrcitive material (FeGaB). This strain is then transferred to the piezoelectric thin-film, and finally the piezoelectric material generates an RF voltage at the output that can be used to power electronic circuitry of the medical implantable device. As mentioned before, the thickness mode of the ME antenna with a resonance frequency of 2.51 GHz is used for energy harvesting because it outperforms the width mode for this application, while the width mode of the smart ME antenna exhibits higher magnetic field sensitivity. The ground ring shown in Fig. 2a forms two symmetrical loops with the center path, where the two ground pads are shorted by wire-bonding and the middle pad is the signal. As it was shown in the previous article by our group[15] the ground ring has a negligible contribution and impact on the ME antenna's performance. This

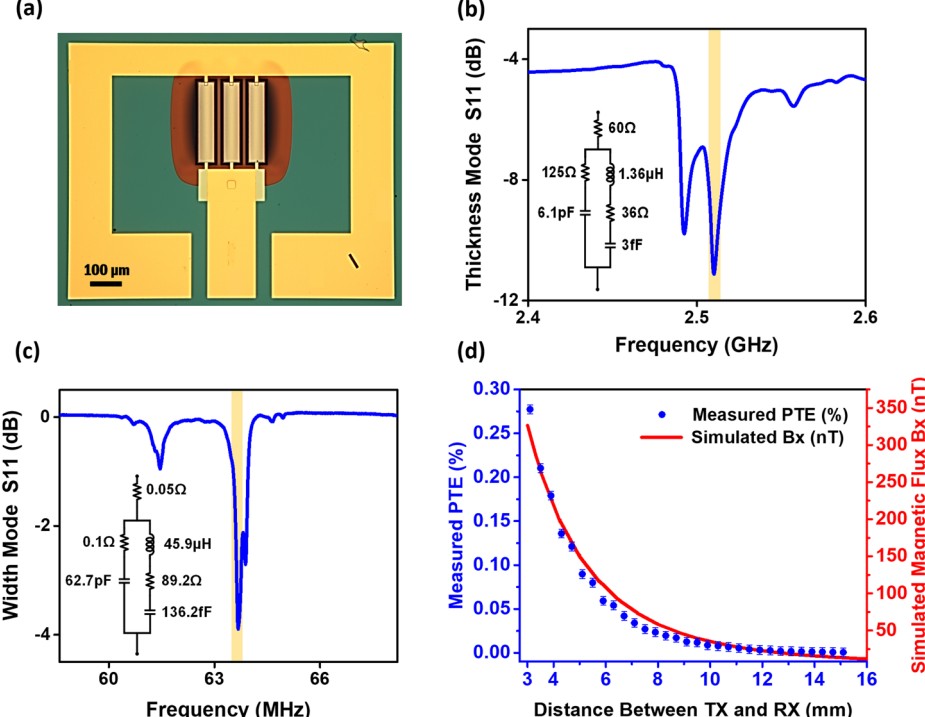

**Fig. 2 ME antenna optical image and measurement results. a** Optical microscope image of a fabricated smart ME antenna; **b** S_11 of the thickness mode of the smart ME antenna showing a resonance frequency at 2.51 GHz; **c** S_11 of the width mode of the smart ME antenna showing a resonance frequency at 63.6 MHz. The insets in (**b**) and (**c**) show the equivalent MBVD circuit model of ME antenna in each mode; **d** The left-axis shows the measured PTE versus distance between ME antenna and Tx coil, and the right axis shows the simulated magnetic flux density along the x-axis (Bx) generated by the Tx coil versus distance from Tx coil.

is mainly because the electromotive force generated by these two loops cancel each other and the net contribution is insignificant.

The diagram of the experimental setup for the energy harvesting measurements is shown in Appendix E, Fig. S5. A single turn $T_X$ coil on an FR4 PCB, shown in Appendix F Fig. S7, is designed to investigate the energy harvesting performance and efficiency of ME antennas. The ME antenna and Tx coil are placed on two 3D printed plastic manipulators in order to precisely control their position. Plastic manipulators, instead of metallic ones, are used to minimize the effect of objects surrounding the Tx coil and their impact on $S_{11}$ of the coil. The manipulators are adjustable along the X, Y, and Z axes, and have a spatial resolution of less than 200 μm, allowing an accurate investigation of impacts of distance and misalignment between the ME antenna and Tx coil. To measure the power transfer efficiency (i.e., the ratio of received power by the ME antenna to transmitted power from the Tx coil), the ME antenna was connected to a power spectrum analyzer, and the Tx coil was connected directly to a VNA under 7 dBm power. The left-axis in Fig. 2d shows the measured PTE versus distance between ME antenna and Tx coil, and right axis shows the simulated magnetic flux density along the x-axis ($\mathbf{B_x}$) generated by the Tx coil versus distance from the Tx coil. The Cartesian coordinate system here is the same as that in Fig. 1b. It is important to note that the ME antenna is sensitive along the x-axis due to the direction of the FeGaB material's easy axis, which is defined based on the applied DC field during the deposition process of this layer. Therefore, the ME antenna only responds to the magnetic fields along the x-axis when generating a voltage. Figure 2d shows that the rate of reduction of the measured PTE and the simulated magnetic flux density along the x-axis ($\mathbf{B_x}$) are matching, and both are decreasing approximately cubically with the distance. The Tx coil was simulated with the COMSOL AC/DC module at 2.51 GHz under an input current of 10 mA, which

corresponds to 7 dBm power (with 50 Ohm resonance condition as in this experiment).

**Misalignment and rotation of the smart ME antenna during energy harvesting**. Figure 3a shows the simulated PTE of a ME antenna in the $XY$ plane at a fixed distance of $z = 8$ mm from Tx coil. In other words, this plot shows PTE versus misalignment of the ME antenna with respect to Tx coil. For this plot, the PTE was measured at 361 points (array of $19 \times 19$ pixels) with a step size of 1.6 mm along both $X$ and $Y$ axes, giving a total measurement range of $14.4 \times 14.4$ mm². The black outline inside the plot identifies the position of the Tx coil within the $XY$ plane. The measurement results show that the PTE, which is proportional to the amplitude of the magnetic flux density ($\mathbf{B_x}$) to which the ME antenna is sensitive, reaches a maximum above the PCB traces along the $Y$ axis, which produces a magnetic flux density along the $X$ axis. Furthermore, the magnetic flux focal size above the longer PCB trace is much larger than the focal point size above the shorter PCB trace. It is noteworthy that the ME antenna was in a fixed position during this measurement, and the Tx coil was shifted along the $X$ and $Y$ axes, which, due to reciprocity theory of electromagnetics, is equivalent to fixing the Tx coil and moving the ME antenna along $X$ and $Y$ axis. Figure 3b shows the COMSOL simulation results for the magnetic flux ($\mathbf{B_x}$) in the $XY$ plane at $z = 8$ mm distance from the coil. The black outline shows the position of the Tx coil within the $XY$ plane. The simulation results confirm the PTE measurements, and they show that the longer PCB trace on the right side of the Tx coil produces an 11× larger magnetic flux focal point than the shorter trace on the left, while this trace is only about 2× longer than the shorter trace.

In biomedical applications, when an implantable device is inserted into the body or brain, inevitably that device will move

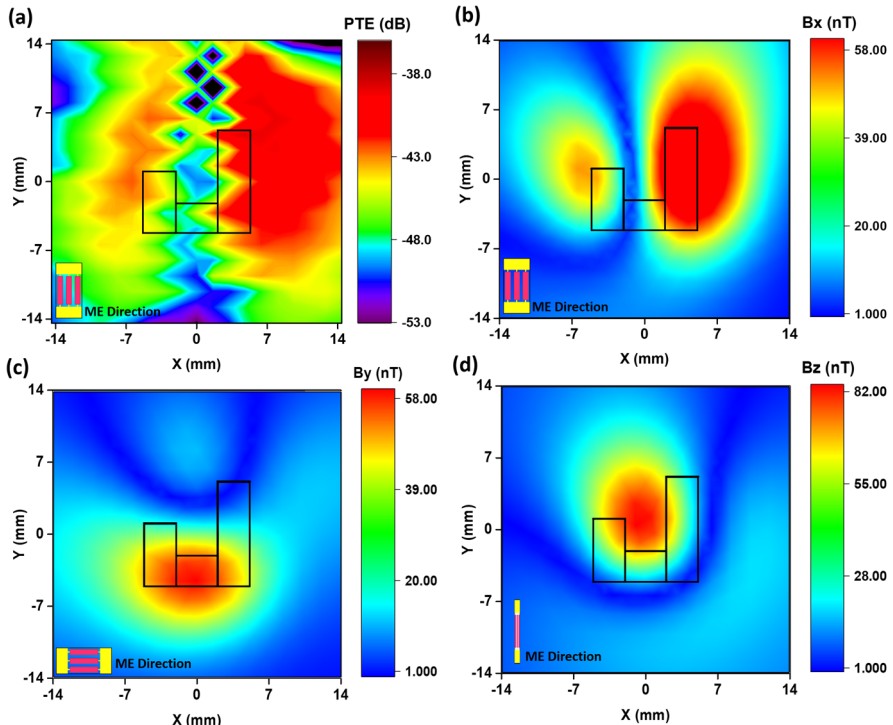

**Fig. 3 COMSOL simulation of the ME antenna and the Tx coil misalignment. a** Measured PTE of the ME antenna in the *XY* plane at a fixed distance of $z = 8$ mm from the Tx coil. The black figure inside the plot shows the coordinate of the Tx coil in the *XY* plane; **b** COMSOL simulation results of magnetic flux (Bx) in the *XY* plane at $z = 8$ mm distance from the coil. The simulation results confirm the PTE measurements and show that the longer PCB trace on the right side of Tx coil produces a much larger (> 11×) magnetic flux focal point than the shorter trace on the left; **c** and **d** show the magnetic flux densities By and Bz in the *XY* plane at $z = 8$ mm distance from the Tx coil, respectively. The magnetic flux By reaches a maximum above the lower copper trace because the current is flowing along the *X* axis in this direction. The magnetic flux Bz is peaking at the center of the Tx coil because the field loops from copper traces are constructively interfering and giving a field stronger than Bx and By.

and rotate after its initial placement. In such a situation with a change of orientation, and considering that the ME antenna is only sensitive to magnetic flux fields along the *X* axis, it is crucial to ensure that the external Tx coil will be able to power the implanted device even if it is oriented towards the Y or Z axes, or any other direction in-between. Simulation results show that the designed Tx coil, or any other rectangular-shaped coil, is able to produce magnetic flux density along almost every direction, including X, Y, Z, XY, etc. This is because the rectangular Tx coil has trace lines along both *X* and *Y* axes, each producing magnetic flux along the Y and X directions, respectively. Moreover, since the Tx coil is a one-turn loop, the magnetic fluxes generated by copper traces constructively interfere at the center of the coil, and produce an even stronger magnetic field along the *Z* axis. Figure 3c and d shows the magnetic flux densities $B_y$ and $B_z$ in the *XY* plane at $z = 8$ mm distance from Tx coil. The magnetic flux By reaches a maximum above the lower copper trace because the current is flowing along X the axis in this direction. The magnetic flux $B_z$ is peaking at the center of the Tx coil because the field loops from copper traces are constructively interfering and giving a field stronger than $B_x$ and By. The magnetic flux distribution along several other directions is available in Fig. S3 in Appendix C. Fig. S3b, for instance, shows the magnetic field along the $y = x$ line—i.e. 45° away from both *x* and *y* axis—in the *XY* plane at z = 8 mm distance from $T_X$ coil. The fields along both *X* and *Y* axes, each with a $\frac{1}{\sqrt{2}}$ factor contribution, produce the magnetic fields along this line.

Based on the discussed results, we can safely conclude that, regardless of how the ME antenna moves and its orientation after implanting the device, by scanning the $T_X$ coil one can align the

peak of magnetic flux to the ME antenna and power the implant. For instance, if the ME antenna is rotated in a way that its sensitive direction is along the *Z* axis, the center of the $T_X$ coil should be aligned to the implant to power the device.

**In-vitro wireless energy harvesting experiment on mice brain tissue.** The discussion in the previous section was on the wireless energy harvesting performance of ME antenna in the air medium. In real biomedical applications, however, it important to take into account the effect of the tissue medium because tissue, when placed between Tx and Rx antennas, can impact the impedance matching of Tx coil and also the magnetic field distribution. In order to investigate the tissue impact on wireless harvesting tests we have completed experiments in which mice tissue was placed between the Tx coil and the ME antenna. The mice tissue was 6 mm thick, as shown in Fig. 4a, and included all the layers including scalp, skull, meninges layers (dura, arachnoid, pia mater), and grey mater. The tissue was placed on top of the Tx coil with 1.6 mm air gap in between, and the ME antenna was precisely aligned on top of the tissue, with 1 mm air gap, to get the maximum power. The total thickness between Tx coil and ME antenna was 8.6 mm. It is notable that when the tissue was placed on top of the Tx coil, the resonance frequency of Tx coil was shifted downward by about 100–300 MHz, which is attributed to the changes of the boundary conditions and also of the impedance seen by Tx coil compared to the air medium. Therefore, the Tx coil was re-matched using an L-match capacitive network after placing the tissue to ensure maximum power is transmitted by the coil.

It turns out that the measured wireless PTE of the system is about 3.1 times higher when the tissue is present compared to air

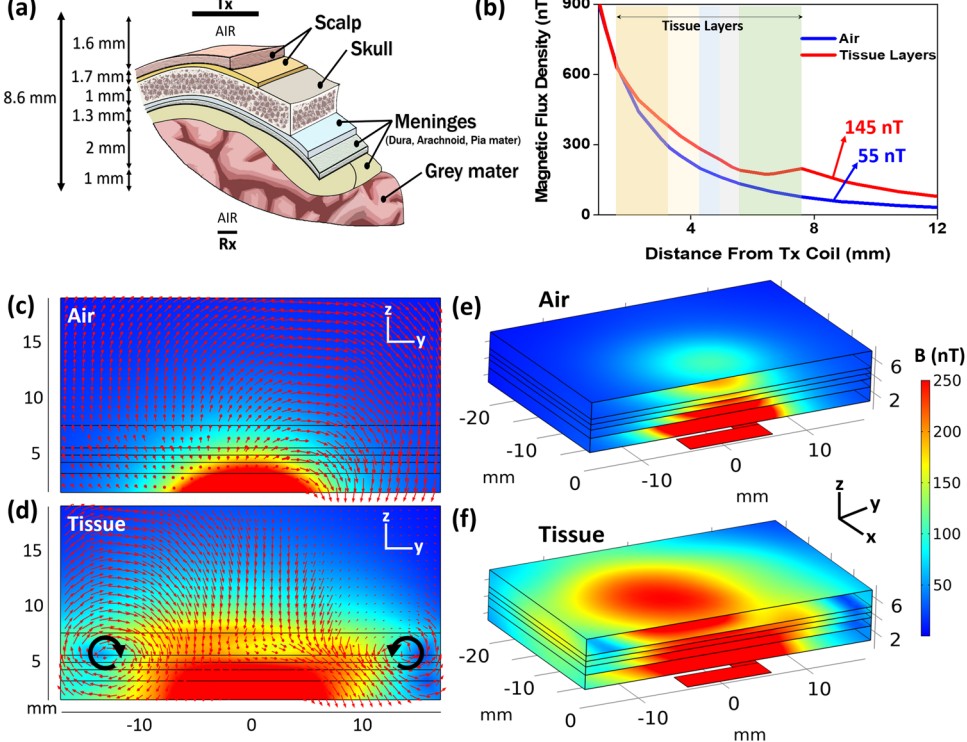

**Fig. 4 Magnetic flux density simulation of the mice brain tissue. a** Schematic of mice tissue layers used in in-vitro energy harvesting experiment; **b** Magnetic flux density vs distance from Tx coil for tissue layers and air medium, showing that the magnetic flux density is stronger when tissue is present. Color bands show the coordinate of different tissue layers, the red one is magnetic flux density in tissue layers and the blue one is in the air; **c, d** simulation results showing the magnetic field distribution and vectors on a cross-section cut of air and tissue mediums, respectively. The black lines show the coordinate of tissue layers. The two vortexes in 6d are due to the magnetic fields generated by the eddy current loops inside the meninges layer, where tissue conductivity is maximum. Tx coil is located at the bottom. The y-axis shows the distance from Tx coil; **e, f** shows the perspective view of magnetic field distribution on a cross-section cut inside air and tissue mediums, respectively. The results in **c, d, e,** and **f** show that the magnetic field distribution in the space changes and is higher when a tissue is present. The color bar scale on the right is valid for all figures.

medium, which was against our initial intuition because the tissue could add some loss to the system and in fact reduce the PTE. In order to shed light on this observation, we simulated and compared the tissue and air mediums using the COMSOL AC/DC module under the same current and frequency conditions as our experiment. Simulation results also matched the experiments and showed that the magnetic flux density at 8.6 mm distance is higher when the tissue is present. Figure 4b shows the magnetic flux density versus the distance from the Tx coil for the air and mice tissue models. The simulation data shows that the magnetic flux density at 8.6 mm is 2.6 times higher when tissue is present compared to air medium. This field enhancement is due to the fact that EM wave is in propagation mode inside the tissue rather than in evanescent near field coupling mode where magnetic field reduces more rapidly. Since the relative permittivity of the tissue is 40–60 times higher than that of air at 2.51 GHz, the EM wave travels slower in the tissue medium and therefore its effective wavelength is shorter. Hence, an EM wave can be propagative even in regions very close to the coil. It is notable that Ho et al. in[19] have observed the same phenomenon where the EM field is stronger when a tissue medium is present. Figure 4c and d shows the magnetic flux distribution and vectors on a cross-section of air and tissue layers, respectively. The black lines show the coordinate of the tissue layers. The magnetic field distribution and vector flows are normal in the air medium, but they are distorted and noticeably changed in the tissue medium. As it is shown, there are two vortexes in the tissue medium at the interface of the meninges layer, where conductivity of the tissue is the highest. The reason for these vortexes is that the Bz magnetic

field created by Tx coil generates eddy current loops inside the tissue layers, particularly in the meninges layers because of their high conductivity, and these eddy currents loops which exist on x-y plane create their own magnetic flux densities. The magnetic fields created by eddy current loops inside the tissue interfere with the Tx coil magnetic fields, creating the vortexes and distorting the magnetic field distribution compared to the air medium. More detailed explanation on the generated eddy currents in the tissue is available in Appendix D. Figure 4e and f shows the 3D view of magnetic field distribution in air and tissue mediums, respectively. It can be observed that the magnetic flux density is higher in the tissue medium, which matches the experimental observation. More details on the electromagnetic simulation of the brain tissue layers are available in Appendix G.

Table 1 compares the ME antenna's energy harvesting performance with that of five other state-of-the-art works selected because of their sub-mm size. These other systems, to the best of our knowledge, are the best published small-size micro-coils. The table compares different systems in terms of PTE, size of energy harvesting element (micro-coil or ME antenna), and the distance between Rx and Tx, as well as the overall energy harvesting performance using the figure of merit (FOM) from[20] that is defined as follows:

$$\text{FOM} = \frac{\eta(\%) \times d^3}{A^{1.5}} \quad (1)$$

where $\eta$ is the PTE as percentage, $d$ is the distance between Rx and Tx in mm, and $A$ is the area of the Rx coil or ME antenna (in

**Table 1 Comparison of the ME antenna's energy harvesting performance with that of five other published micro-coils with sub-mm size.**

| Ref. | Rx Area (mm²) | RX TYPE | Tx-Rx Distance (mm) | Wireless System | Medium | PTE(%)=$\eta_{Tx}$*Path Loss*$\eta_{Rx}$ | FOM |
|---|---|---|---|---|---|---|---|
| Dukju[28] | 0.79 | Off-chip coil | 12 | 2-coil | Tissue (beef) | 0.56 [200 MHz, PDL = 224 µW] | 1378 |
| Khalifa[11] | 0.09 | On-chip coil | 6.6 | 2-coil | Tissue (lamb) | 0.01* | 106 |
| Liuqing[29] | 0.01 | On-chip coil | 0.5 | 3-coil | Air | 0.12 [2 GHz] | 12 |
| Nai-Chung Kuo[30] | 0.01 | On-chip coil | 2.2 | 2-coil | Air | 0.0079* [4.8GHz, PDL = 100 µW] | 84 |
| This work (Smart ME antenna) | 0.044 (250 × 174 µm²) | ME antenna | 8.6 | 1-coil + ME antenna | Mice Head Layers | 0.054 [2.51GHz] | 3721 |

*Simulation. The formula in Eq. (1) is used to calculate the figure of merit. This formula considers the area of the receiver, the distance between Rx and Tx, and the wireless power transfer efficiency..

this work, $250 \times 174\,\mu m^2$ is the area of the ME resonators) in $mm^2$. As it can be observed in Table 1, the FOM of the ME antenna is better than that of other mm-scale coils (row 1 in table) and significantly superior to other µm-scale coils (row 2–5 in the table).

**Magnetic field sensing with smart ME antenna.** Ultracompact magnetic sensors are of use in various applications such as biomedical imaging, unmanned autonomous systems, and geophysical studies. For biomedical applications in particular, such as recording magnetic fields of brain neural activities or mapping magnetic patterns of the heartbeat, an ideal magnetometer should have a high spatial and temporal resolution to reduce invasiveness and to be able to record fast-changing magnetic signals. Moreover, the magnetometer should operate well in an un-shielded and room-temperature environment. The vast majority of existing magnetometers lack at least one of the mentioned criteria, and cannot be used for biomedical applications. In this section, we will discuss the underlying concept of magnetic field sensing using the proposed smart ME antenna, and will also assess the performance of this device for sensing sub-nT magnetic fields.

The magnetic sensing functionality of an ME antenna is based on a non-linear ME modulation technique[21], where the external magnetic field is modulated on the RF carrier signal that is applied to the ME antenna. As mentioned before, the in-plane mode of a smart ME antenna is used for magnetic field sensing because, based on our observations, this mode shows a significantly better performance for this purpose than the thickness mode. The three parallel ME elements in the smart ME antenna each consist of a FeGaB and AlN heterostructure, where the former acts as magnetostrictive layer and the latter acts as piezoelectric layer. The coupling between magnetic, electric, elastic, and thermal parameters of this FeGaB/AlN heterostructure can be expressed using a thermodynamic approach[22]. Assuming that the temperature is constant in our experiment, and since the external magnetic and electric fields, as well as the external elastic force, are all independent variables, Gibbs free energy $G(T, \sigma, H, E)$ for multiferroic materials can be used as thermodynamic potential[22] and, accordingly, the total strain (x) of the system can be expressed as below:

$$x_{ij} = s_{ijkl}\sigma_{kl} + d_{ijk}^{e}E_k + d_{ijk}^{m}H_k + \pi_{ijkl}E_kH_l \qquad (2)$$

where $s_{ijkl}$ is the elastic compliance tensor, $\sigma_{kl}$ is the mechanical external applied stress tensor, $d_{ijk}^{e}$ is the piezoelectric coefficient tensor of AlN thin-film, $\mathbf{E}$ is the external electric field, $d_{ijk}^{m}$ is the piezomagnetic coefficient tensor of FeGaB layer, $\mathbf{H}$ is the external applied magnetic field, and $\pi_{ijkl}$ is the piezo-coupling constant tensor. There is no external applied stress in our measurements, and therefore the first term, $s_{ijkl}\sigma_{kl}$, which is the strain due to external mechanical stress, is almost zero and can be neglected. Thus, Eq. (2) can be written as below:

$$x_{ij} = d_{ijk}^{e}E_k + d_{ijk}^{m}H_k + \pi_{ijkl}E_kH_l \qquad (3)$$

where the first term is the piezoelectric strain induced by the external electric field (RF voltage) applied to AlN thin-film, the second term is the piezomagnetic strain due to external magnetic field (the field that is being recorded), and the third term, which is the ME modulation term, is the piezo-coupling strain due to the mutual piezoelectric/piezomagnetic effects or, in other word, the strain-mediated ME effect in the AlN/FeGaB heterostructure. The external applied electric field can be written as $E = \frac{V}{t_{AlN}}$, where $V = V_c \cos(\omega_c t)$ and $t_{AlN}$ are the applied RF voltage to and thickness of the AlN layer, respectively; and the external magnetic field can be expressed as $H = H_m \cos(\omega_m t)$. It is notable that $V_c$

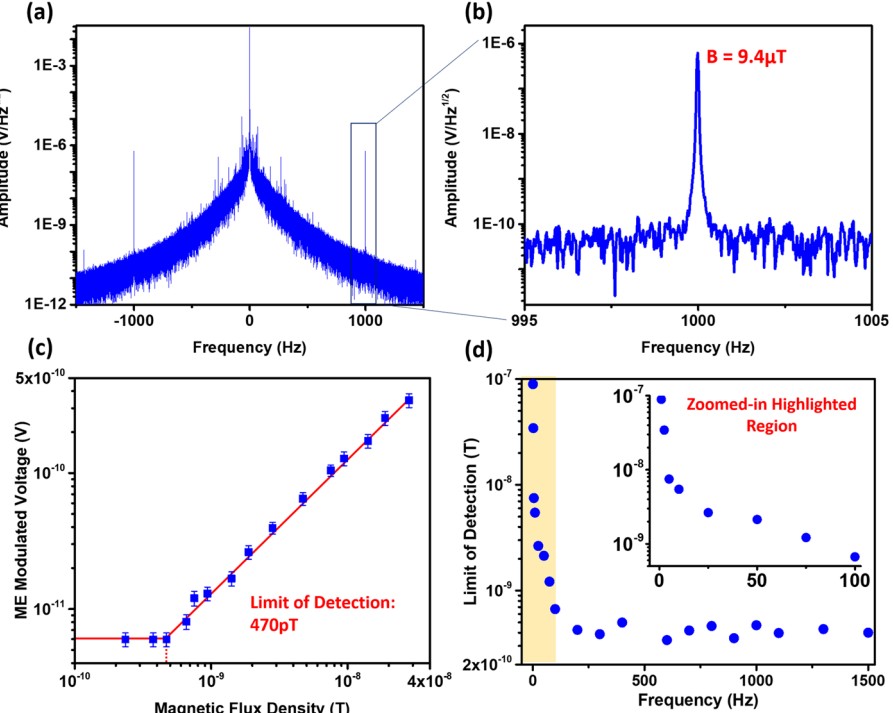

**Fig. 5 ME antenna magnetic sensing measurement results. a** Power spectrum of the reflected signal from the ME antenna after demodulation and low-pass filtering. The plot shows both modulation components of $\omega_c - \omega_m$ and $\omega_c + \omega_m$. The carrier signal is located at the center and has an amplitude of $2.9 \times 10^{-2} \frac{V}{\sqrt{Hz}}$. The noise figure decreases significantly with increasing frequency—which is because of the low-pass filtering and the $1/f$ electronics noise reduction—improving the SNR of the sensor at higher frequencies; **b** The zoomed-in plot shows the modulation signal at $1kHz$ with an amplitude of $6.2 \times 10^{-7} \frac{V}{\sqrt{Hz}}$, which corresponds to an external magnetic field of $H_m = 9.4\mu T$; **c** ME modulated voltage versus amplitude of the external magnetic field $H_m$ at $1kHz$ frequency, showing a LOD of 470 pT; **d** LOD of ME antenna versus frequency of external alternating magnetic field $H_m$. The LOD of the sensors improves with increasing the frequency because $1/f$ electronics noise is strong at very low frequencies. At frequencies higher than $100Hz$, both noise and ME modulation voltage decrease with almost the same rate, which is why the LOD, or i.e. the SNR, becomes almost flat. The inset plot shows the zoomed-in highlighted region (frequencies lower than 100 Hz).

and $\omega_c$ are the amplitude and frequency of the carrier electric signal applied to ME antenna respectively, and $\omega_c$ is equal to the resonance frequency of the CMR mode; $H_m$ and $\omega_m$ are the amplitude and frequency of external the alternating magnetic field. The third term in Eq. (3), which is the strain due to ME effect, can be simplified as follows to demonstrate how the modulation phenomenon occurs:

$$x_{ME} \propto \frac{1}{t_{AlN}} V_c \cos(\omega_c t) \times H_m \cos(\omega_m t) \qquad (4)$$

The above equation expresses the modulation between carrier voltage with frequency of $\omega_c$ and modulation field with frequency of $\omega_m$. The modulation frequencies arising from the strain-mediated ME effect expressed in this equation are $\omega_c - \omega_m$ and $\omega_c + \omega_m$, and the amplitude of these modulation components is linearly proportional to the amplitude of external magnetic field $H_m$, which implies the usefulness as a sensor. The linearity of the modulation components will be discussed later in this section based on Fig. 5c. Furthermore, since the low-frequency external alternating magnetic field is modulated on a high-frequency signal (at the CMR resonance frequency) and therefore its information is transferred to a higher frequency bands, the modulated component has more immunity from environmental electromagnetic, thermal, and mechanical noises that are typically all more severe in lower frequency ranges where they can easily degrade the signal-to-noise ratio (SNR) of the sensor. For instance, the work in[21] has demonstrated that a magnetoelectric modulation technique can improve the SNR of the sensor up to 72 times compared to traditional DC-biased and unmodulated

sensors. It is notable that even though the mentioned work was completed on a very large magnetoelectric heterostructure, 28 mm in length, the concept, and the modulation technique are similar to our scenario.

The diagram of experimental setup for magnetic sensing using an ME antenna is shown in Appendix E, Fig. S6. The ME antenna is excited using a lock-in amplifier through a directional coupler, and is operating with a 140 mV amplitude signal at its width resonance frequency of 63.6 MHz. The other port of the directional coupler, which is connected to the input of the lock-in amplifier for demodulation and post-processing, carries the reflected signal from ME antenna and contains the strain-mediated ME response of the FeGaB/AlN heterostructure. Note that the measurements were performed using the lock-in amplifier's built-in filter set to a cutoff frequency of 89 Hz. The digital filter in the lock-in amplifier is a 4th-order low-pass that is applied to the digitized signal after mixing the incoming signal and the reference. The displayed noise floor level is subject to the settings of this discrete-time RC filter in the lock-in amplifier, which is a moving average filter that significantly reduces the noise level. In the shown measurement results, the filter suppresses the modulated signal and the noise level by almost 66 dB, while improving the SNR of the processed signal by a factor of 12 times according to our experiments. When excited (by RF voltage), the ME antenna is in presence of an external alternating magnetic field, and its reflected signal carries modulated components $\omega_c - \omega_m$ and $\omega_c + \omega_m$, the amplitude of which is an indicator of the external magnetic field. A 37 Oe DC bias field provided by the larger Helmholtz coil (black color)

is applied perpendicular to the length of the resonator in order to maximize the magnetoelectric coefficient and sensitivity. We have investigated the impact of the external DC bias field on the sensitivity of the sensor and applied DC fields from 0 to 70 Oe. The sensor performance and sensitivity were at their peak under the external DC bias field of 37 Oe. The reflected signal from the ME antenna is demodulated and low-pass-filtered after going through the input of the lock-in amplifier, and then its FFT is analyzed on a computer. Figure 5a displays the power spectrum of the reflected signal from the ME antenna after demodulation and filtering. The plot shows both modulation components of $\omega_c - \omega_m$ and $\omega_c + \omega_m$ on two sides of the carrier signal that is located at the center and has an amplitude of $2.9 \times 10^{-2} \frac{V}{\sqrt{Hz}}$. As it is shown, the noise figure decreases significantly with increasing the frequency—which is because of the low-pass filtering and the low-frequency electronics noise reduction—improving the SNR of the sensor at higher frequencies. Figure 5b shows the zoomed-in modulation signal of $\omega_c + \omega_m$ at 1 kHz with an amplitude of $6.2 \times 10^{-7} \frac{V}{\sqrt{Hz}}$ that corresponds to an external magnetic field of $H_m = 9.4 \mu T$. Figure 5c shows the ME modulated voltage versus amplitude of the external magnetic field $H_m$ at 1kHz frequency. As discussed before in Eq. (4), the amplitude of the ME modulation signal changes linearly with external alternating magnetic field $H_m$, making the device suitable to be used as a sensor. The plot shows that the limit of detection (LOD), or minimum detectable magnetic field, of the sensor is 470 pT. The ME modulation signal is at the same level of noise under magnetic fields smaller than LOD and therefore is lost in the noise. Figure 5d shows the LOD of ME antenna versus frequency of external alternating magnetic field $H_m$. As it was shown in Fig. 5a, the voltage of the post-processed reflected signal decreases with increasing the frequency. At frequencies under 100 Hz, the noise level decreases faster than the ME modulation peak because of the large electronics noises at lower frequencies, which implies that the SNR of the ME antenna, and therefore the LOD, improves with increasing frequency as demonstrated in Fig. 5d. For modulation frequencies above 100 Hz, on the other hand, both the noise and ME modulation signal decrease almost at the same rate, which is why the LOD of the sensor, i.e. the SNR, becomes flat. The inset plot in Fig. 5d shows the zoomed-in highlighted region (frequencies lower than 100 Hz), revealing a rapid change of the LOD at frequencies close to DC. Recent results in[23] show that the amplitude of the neural magnetic fields, for in-vivo measurement where the sensor is 10–100 s of μm away from neural ensembles, is as high as 1–10 nT. In another work from Wikswo et al.[24], the authors reported a neural magnetic field with a frequency of approximately 1 kHz and an amplitude of 120 pT at 1.3 mm from the nerve. This value at this distance approximately agrees with the data in[23] where the modelling results in 126 pT at 1 mm, 1.3 nT at 100 μm, and 2.3 nT inside the neuronal ensemble. Therefore, the magnetic field measurements in these two works suggest that our ultra-compact smart ME antenna with a limit of detection of <470 pT has the capability to record neuronal magnetic fields.

**Simultaneous wireless energy harvesting and magnetic field sensing**. The wireless energy harvesting and magnetic field sensing experiments in previous sections were carried out separately and independently. In this section we will discuss the performance of the smart ME antenna when it is harvesting EM waves and sensing external magnetic fields simultaneously. The diagram of the experimental setup used for this test is shown in Fig. S8 in Appendix H. In this test we have merged the experimental setups used in energy harvesting and magnetic field sensing measurements shown in the Figs. S5 and S6 in Appendix E. Here the ME

antenna is in presence of four external forces simultaneously: (1) a 2.51 GHz EM wave generated by the Tx coil used for the energy harvesting purpose; (2) a 63.6 MHz RF signal from the lock-in amplifier in order to enable the magnetic sensing mode of the device; (3) a 37 Oe DC bias field generated by the large-size (black) Helmholtz coil in order to maximize the sensitivity and magnetoelectric coefficient of the sensor; (4) a low-frequency external magnetic field generated by the small-size (blue) Helmholtz coil as a test field. The ME antenna is fixed on a holder at the center of the Helmholtz coils, but the Tx coil is mounted on a plastic manipulator in order to be able to adjust its XYZ coordinates with respect to the ME antenna. Via an SMA T-splitter the ME antenna is connected to both a power spectrum analyzer to read the harvested 2.51 GHz power and the lock-in amplifier via a coupler to monitor the magnetic field sensing performance of the device. Energy harvesting and magnetic sensing operation principles are similar to those discussed in previous sections and the main difference here is the use of a T-splitter to simultaneously monitor both functionalities of the ME antenna.

We performed the simultaneous energy harvesting and magnetic field sensing experiment in three different scenarios: (1) the device was operating only at its magnetic field sensing mode where the 2.51 GHz Tx coil was off and there was no splitter in the circuit; (2) the device was operating at its energy harvesting mode where there was no 63.6 MHz excitation, both the Helmholtz coils were off, and the coupler and splitter were disconnected; (3) the device was operating at both the energy harvesting and magnetic sensing mode where the 63.6 MHz excitation and the 2.51 GHz Tx coil as well as the DC and AC Helmholtz coils were all enabled. The coupler and the splitter were also added to the circuit, as shown in the Fig. S8 in Appendix H. For the energy harvesting tests we recorded the received power by the ME antenna at 2.51 GHz, and for the magnetic field sensing we recorded the ME modulation peak voltages, which are the most important parameter showing the performance of the sensor. The results from the simultaneous experiment showed that the received energy harvesting power at 2.51 GHz was 29% lower compared to the case where only energy harvesting mode was on and there was no splitter in the circuit. Moreover, the magnetic sensing data from the simultaneous experiment showed that the ME modulation voltages were 45% lower compared to the case where only magnetic field sensing mode was on and there was no splitter in the circuit. The observed noise floor was constant and unchanged in all scenarios. We realized that if we keep the splitter in the circuit and turn off the 2.51 GHz Tx coil we observe the same 45% reduction in the ME modulation voltage, and if we turn off the 63.6 MHz RF excitation as well as DC and AC test fields we still have 29% reduction in the harvested power. This observation implies that the act of enabling both the frequency modes of the antenna simultaneously does not degrade the performance of the device—and that the observed signal reduction is solely associated to the presence of the splitter in the circuit which divides the harvested power and ME modulated signal into two lines. The fact that the smart ME antenna can simultaneously operate at both its modes without interference and performance degradation is reasonable given the large difference in the resonance frequencies. It is notable that the difference in the percentage of the reduced energy harvesting power (29%) and the ME modulation voltage (45%) is due to the difference in the input impedance of the two lines after the splitter. The reduced percentage in the ME modulation voltage was consistent at different frequencies of external test magnetic field from 0 to 1500 Hz and was always about 45%. It is noteworthy that even after reduced PTE and magnetic field sensing performance in the simultaneous mode, the wireless energy harvesting FOM of the ME antenna is still

higher than that of other works in the Table 1—And the magnetic field LOD of the sensing mode is still enough to detect the magnetic neural amplitudes mentioned in the[23] and[24].

## Discussion

Future works on this device will include integration with an on-chip integrated circuit (IC) design and the packaging process. The IC design will fully enable the energy harvesting and wireless data transmission features—and the packaging, encapsulation process will protect the ME antenna and IC components against the moisture ingress and prolong the lifetime of the device. The ME antenna fabrication process is CMOS compatible and can be integrated to the IC blocks. The packaging process of the ME antenna, however, could be a challenging and delicate procedure given that the resonator is released and must be vacuum-encapsulated for an optimum resonator performance. This process could be eased by using solidly mounted resonators (SMR)—which are fabricated directly on the substrate and do not need to be released—instead of current resonator design.

We have explored the data transmission via an IC design technique. The proposed method is to synthesize a local oscillator (LO) using the ME antenna operating at its CMR resonance frequency, and then to use this LO to apply a sinusoidal-like excitation current at the ME antenna[25]. This produces an amplitude-modulated output when there is a change in the magnetic field due to neuronal activity. The diagram of the proposed IC design is shown in Appendix I, where the device with the ME antenna will require circuitry for energy harvesting at the FBAR resonance frequency, as well as a driver circuit that generates a carrier signal at the CMR resonance frequency for transmission of the sensed information[26,27]. The transmitted magnetically amplitude-modulated signal has to be amplified and down-converted by the external RF receiver front-end, and further processed with digital filtering to extract the signal components that carry the sensed magnetic field information. Using the proposed IC design we can also align the external Tx coil to the implanted device in order to get the highest wireless power transfer efficiency. The device generates the largest power when the ME antenna is at the most optimum angle for the energy harvesting, which in turn it transmits back the strongest amplitude-modulated signal. In other words, based on the strength of the externally received amplitude-modulated signal we can align the Tx coil to the implanted device to achieve the highest power transfer efficiency.

We have reported an ultra-compact dual-band smart NEMS ME antenna that can be used for biomedical applications, in particular for implantable medical devices. The proposed dual-band ME antenna with a size of $250 \times 174\ \mu m^2$ has two acoustic resonance frequencies at 2.51 GHz and 63.6 MHz, where the former mode is used for wireless RF energy harvesting and the latter for low frequency magnetic field sensing, which can be used for neural recording. The wireless power transfer efficiency of the smart ME antenna is 1–2 orders of magnitude better than that of any other reported miniaturized micro-coils to date. The improved PTE allows the implantable device to operate at higher depth inside the body while adhering to the SAR limit set by the FCC. It was also shown that the ME antenna can be efficiently powered using an external Tx coil even if the device is misaligned or rotated after implantation. In addition, the smart ME antenna's magnetic sensing mode shows an ultra-low limit of detection of less than 470 pT, which can be used for neuronal magnetic field sensing.

## Data availability

The data that support the findings of this study are available from the corresponding author upon request.

## Animals

All experiments were approved and monitored by the Massachusetts General Hospital (MGH) Institutional Animal Care and Use Committee (protocol no. 2015N000107). Adult Sprague Dawley rats were anesthetized using isoflurane gas and quickly followed by decapitation using a commercially available guillotine. The brain, dura, skull, and scalp were then removed.

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

## Acknowledgements

This work is partially supported by the NIH Award UF1NS107694 and by the NSF TANMS ERC Award 1160504.

## Author contributions

M.Z. initiated and designed the experiments, performed biological and energy harvesting modeling, and led the measurements under the supervision of N.X.S., M.O., A.S., and S.C.; M.N. and A.K. assisted with the energy harvesting measurements and data interpretation; H.C., X.L., and N.S. assisted with the fabrication of the ME devices; A.M. helped in the magnetic field sensing experiments; H.L. assisted with the simulation of the strain and displacement distribution in FeGaB and AlN thin-films; A.R., C.D., Z.X., A.M., I.M.R., G.J., N.M., and D.D. helped with the test setup. All authors discussed the results.

## Competing interests

The authors declare no competing interests.
