## [Peer Review File · Nature Communications]

Reviewer #1:

Remarks to the Author:

Review of NCOMMS-20-15513

The authors report on ultra-compact wireless implantable medical devices that are in great demand for healthcare applications, in particular for neural recording and stimulation. In this manuscript they report on a nanoelectromechanical systems magnetolectric antenna with a size of $250 \times 174 \mu\text{m}^2$ that can efficiently perform wireless energy harvesting and sense magnetic fields arising from neural activities. This is in general a very interesting topic that is worth be published in Nature Communications.

However, the structure of the manuscript needs to be revised significantly:

- 1.) The three-part nature of the paper is very confusing reading the article, it is unclear what is being done in one section and why. I only understood the motivations reading the conclusion.
- 2.) Why exactly this three resonator design is chosen is not meaningfully explained. The term NPR (nano plate resonator) is used, but the specified lateral dimensions ($250 \times 50 \mu\text{m}$) do not give any legitimation for the prefix nano. What should happen to the harvested energy in the body has also remained somehow unclear to me as e.g. no autonomous sensor system is described.
- 3.) Fig. 3, PCB transmitter antenna is state of the art, the part is far too long for this journal.

Furthermore, a number of questions related to the experiments presented should be answered before publication:

- 1.) In section 2 the specification of a thickness of the resonator or the question of whether there is a remainder of the substrate under the layer stack is not addressed, but seems relevant to me.
- 2.) Please correct the wording (line 125): no peaks are seen, three principal dips are visible in each plot of Fig. 2 b and c.
- 3.) Please provide information about the used network analyser (line 130)
- 4.) In sections 2-1 and 2-2 there is no schematic illustration of the measurement setup. Fig. 4 would have to refer to the receiving antenna (or the ME resonator) and be plotted as directional characteristic 3D - then it would be much easier to understand.
- 5.) In section 2-3, quite unrealistic distances (all in the sub-cm range) are used and the modeling is simply not explained in more detail. Here a very complex field is broken down on two and a half pages.
- 6.) The FOM (equation 1) that is introduced is very simplified, knowing the numerical values of the parameters is assumed, A remains unclear due to the unknown thickness of the resonator. More importantly, d is cubic and is treated here as a simple scalar, but is usually strongly anisotropic for any

antenna with directional characteristics.

- 7.) Eq. (4) is set up, but no further reference is made, and the introduced x_{ME} never occurs again. The role of the bias field of exactly 37 Oe is not discussed further, it is only mentioned once. Unusual to put it along the short axis.
- 8.) Fig. 6 is not so easy to obtain with the measuring device mentioned in the supplement, the plot (a) shows an incredibly large dynamic range of over 200 dB (corresponding to 10 orders of magnitude), the data sheet confesses a dynamic range of 100 dB (briefly above 6 orders of magnitude). Furthermore, the noise falls considerably faster than $1/f$.
- 9.) The noise carpet in Fig. 6 (a) and (b) is two orders of magnitude below the best case input noise voltage of $4 \text{ nV} / \sqrt{\text{Hz}}$ specified by the same measuring device.
- 10.) If one calculates the sensitivity (i.e. slope) shown in Fig. 6 (c) one receives only about 10 mV/T (a standard hall sensor has about 10 V/T), please comment. Furthermore, the sensitivity of 10 mV/T and the noise density of $40 \text{ pV}/\sqrt{\text{Hz}}$ would result in a LOD of only $4 \text{ nT}/\sqrt{\text{Hz}}$.
- 11.) I have also doubts about the given noise levels: Voltage densities in the $\text{pV}/\sqrt{\text{Hz}}$ -range seem to be very unrealistic to me. In a 50 ohm system, 30 or $40 \text{ pV}/\sqrt{\text{Hz}}$ corresponds to at a temperature of less than 1 K.
- 12.) The LOD in Fig. 6 (d) is shown to be flat between 250 Hz and 1500 Hz. The noise (taken from Fig. 6 (a)) increases about three orders from 1500 Hz to 250 Hz (from 40 pV (at 1500 Hz) to about 40 nV (at 250 Hz)), please comment.

Reviewer #2:

Remarks to the Author:

The work is devoted to the investigation of the ultra-compact dual-band Smart NEMS ME antennas for simultaneous wireless energy harvesting and magnetic field sensing. The Authors applied the original method and combined the systems of wireless energy harvesting and measuring of weak magnetic fields in one ME antenna. They used the operation of the ME antenna in two modes (thickness and width) at two frequencies of 2, 51 GHz to collect energy and 63.6 MHz for measuring the magnetic field in neural activities. They examined the issue of the disorientation of Tx and Px in the operating mode and presented experimental data on energy extraction in the study of the brain of mice. The ME antenna developed has shown high results; in size and sensitivity it exceeds the known similar systems on micro- coils. But the Authors need to answer a number of questions.

1. How justified is the choice of frequencies for the thickness and the width modes?
2. How to improve the signal spectrum in Figure 2?
3. How is the area size Rx obtained in Table 1?

The reviewer believes that after answering the questions posed, the article can be published in the Nature Communications

Reviewer #3:

Remarks to the Author:

This manuscript by Sun and colleagues describes the latest developments in their NEMS magnetoelectric (ME) antennas for bioelectronic devices with important new capabilities. Specifically they describe a miniature device with wireless energy harvesting capability and impressive magnetic field sensing ability. The major interest in ME antennas is their ability to achieve good power densities even when the devices are made very small. This is an important property for implanted bioelectronics because small form factor devices are less invasive and enable potentially many independent devices in a small area of tissue. An exciting aspect of this paper is the fact that the same ME antenna used for power harvesting can also be used for magnetic field sensing. Magnetic field sensing is one important method to track activity of electrogenic cells like those in the brain.

Despite the important impact of the advancements discussed by the authors, the manuscript in its current form falls well short of the claims in a few important ways. These are summarized as major points below and followed by minor technical points that should be addressed in any potential revision.

Major Points:

- No simultaneous energy harvesting and sensing: Most importantly, the title suggests "Simultaneous energy harvesting and magnetic field sensing;" however, there appears to be no demonstration of simultaneous operation of power harvesting and magnetic field sensing. The details of the magnetic sensing experiment are not clearly described, but it appears that for the sensing experiments the device is powered via direct electrical connection. The same antenna appears capable of sensing and power harvesting, but there may be difficulty in performing both functions simultaneously and certainly needs to be demonstrated if the authors are to use their

current title.

I would suggest the authors significantly revise their claims and the title to better match what they actually show: the same antenna can function to sense magnetic fields and harvest data - although they do not show the ability to do so simultaneously. Else they should add experiments that clearly show simultaneous power and sensing.

- No uplink to communicate data to the Tx/Rx: As a sensor for an implantable bioelectronic device it is critical to communicate the sensed data back to the Rx. This is a critical element for 1) magnetic sensing, and 2) aligning the transmitter to device (as described on line 222-226).

The authors should clearly identify this need, and discuss potential solutions. Without this element the manuscript is misleading in the sense that much of the capabilities described could not be implemented in any practical way without a significant engineering advance.

The authors should clearly state the shortcomings of their current device and need for future work.

- Well short of in vivo practicality: All of the shortcomings would come through if the authors would perform an in vivo experiment. From what I can tell from the manuscript they are many missing pieces before that could happen. I do not necessarily think they need to complete these experiments before publishing, but the elements that need to be developed before implementation should be clearly stated (e.g. uplink, packaging, angular tolerance etc.). Without this clear and open discussion the manuscript is highly misleading.

Minor points:

Line 182-184, Fig 3-c: The device has a low output power - I think it's lower than the power required for stimulation / IC powering-, what applications could it be used for?

Fig 6, Fig S6, line 348-352: For the testing setup of the magnetic field sensing, a directional coupler was used to carry the reflected signal from the ME antenna that contains the strain-mediated ME response, how the setup will change for the implanted device?. Also, the ME-Modulated voltage (Fig 6-c) was extremely small, -if not carried by the coupler- how to ensure robust operation of the film?

Line 110-114: The formula used to calculate the resonance frequency -for both the width and thickness modes- depends only on the properties of the piezoelectric material, however, in other literature it is a function of both the piezoelectric and the magnetostrictive properties.

Fig 6-a: What does the amplitude of the carrier component represent?

Dear Nature Communications' Reviewers,

The authors appreciate that you have dedicated the time and effort to provide insightful feedback on ways to strengthen our paper. Thus, it is with great pleasure that the authors resubmit our article for further consideration. The authors have incorporated changes that reflect the detailed suggestions you have graciously provided. The authors also hope that our edits and the responses the authors provide below satisfactorily address all the issues and concerns you have noted.

We sincerely apologize for the delayed submission of this revision, which happened due to the lab closure throughout the summer that was caused by the COVID-19.

The responses for Reviewer 1 are written in blue color, the responses for Reviewer 2 are written in green color, and the response for Reviewer 3 are written in red color in the revised manuscript. To facilitate your review of our revisions, the following is a point-by-point response to the questions and comments.

Reviewer 1 (Written in blue color in revised manuscript and supplementary materials)

Part 1: Major Comments

The authors report on ultra-compact wireless implantable medical devices that are in great demand for healthcare applications, in particular for neural recording and stimulation. In this manuscript they report on a nanoelectromechanical systems magnetolectric antenna with a size of $250 \times 174 \mu\text{m}^2$ that can efficiently perform wireless energy harvesting and sense magnetic fields arising from neural activities. This is in general a very interesting topic that is worth be published in Nature Communications.

However, the structure of the manuscript needs to be revised significantly:

- 1) The three-part nature of the paper is very confusing reading the article, it is unclear what is being done in one section and why. I only understood the motivations reading the conclusion.**

Thanks for your comment. We understand that the current structure is different from the conventional Results and Discussion structure. The main reason is that this paper discusses two different topics and functionalities (energy harvesting and magnetic field sensing) using the same ME antenna, which makes it a bit challenging to condense the

materials into the conventional format. That is why we thought this structure would be more suitable and make more sense. We believe it would be harder for the reader to readily find her/his desired information on each topic in the conventional structure.

The paper—as the title, abstract, intro, and conclusion suggest—is on a novel, compact ME antenna for **wireless energy harvesting** and **magnetic field sensing**. We have revised the transitions to make the structure and what is being discussed in each section more clear to the reader. The revised sections are as follow:

- 1- Introduction
- 2- Design and Characteristics of Smart ME Antenna
- 3- Wireless Energy Harvesting with Smart ME Antenna
- 4- Magnetic Field Sensing with Smart ME Antenna
- 5- Simultaneous Wireless Energy Harvesting and Magnetic Field Sensing
- 6- Conclusion

Where section 2 discusses the operational concept, geometry, and frequency modes of the ME antenna. Sections 3 and 4 discuss energy harvesting and magnetic field sensing principles and performance of ME antenna, respectively. Section 5 discusses the simultaneous energy harvesting and magnetic field sensing using the smart ME antenna. We hope that the current structure and the motivation for each section are more clear.

2) Why exactly this three resonator design is chosen is not meaningfully explained. The term NPR (nano plate resonator) is used, but the specified lateral dimensions (250x50 μm) do not give any legitimation for the prefix nano. What should happen to the harvested energy in the body has also remained somehow unclear to me as e.g. no autonomous sensor system is described.

Thanks for the comment. Regarding the three resonators selection, we have added this statement to the manuscript to make it more clear:

“We have designed ME antennas with one to seven number of parallel resonators. Our experiments have shown that the energy harvesting and magnetic field sensing performance of the ME antenna enhance by increasing the number of parallel resonators. This is due to the increased effective area of the antenna that is in presence of external magnetic field. Even though using an antenna with seven parallel resonators will have a better performance, the size of antenna will consequently increase, too—which is not desirable for our target application. Thus, in this project we have selected an antenna with three parallel resonators to have a relatively small size and good energy harvesting and magnetic field sensing performance.”

Regarding the term NPR, we have changed the name to contour mode resonator (CMR) in order to avoid confusion. It is noteworthy that some works published by other groups with comparable length, width, and thickness have used the term NPR in their manuscript. These works have a slightly thinner thickness than our device has. Below are some of those works.

1- A resonator with $75 \times 200 \mu\text{m}^2$ size:

Hui, Yu, Juan Sebastian Gomez-Diaz, Zhenyun Qian, Andrea Alu, and Matteo Rinaldi. "Plasmonic piezoelectric nanomechanical resonator for spectrally selective infrared sensing." *Nature communications* 7, no. 1 (2016): 1-9.

Qian, Zhenyun, Yu Hui, Fangze Liu, Sungho Kang, Swastik Kar, and Matteo Rinaldi. "Graphene–aluminum nitride NEMS resonant infrared detector." *Microsystems & nanoengineering* 2, no. 1 (2016): 1-7.

2- A resonator with $50 \times 196 \mu\text{m}^2$ size:

Qian, Zhiming, Y. Hui, F. Liu, S. Kar, and M. Rinaldi. "245 MHz graphene-aluminum nitride nano plate resonator." In *2013 Transducers & Eurosensors XXVII: The 17th International Conference on Solid-State Sensors, Actuators and Microsystems (TRANSDUCERS & EUROSENSORS XXVII)*, pp. 2005-2008. IEEE, 2013.

3- Other articles that have used the term nanoplate resonators

Qian, Zhenyun, Yu Hui, and Matteo Rinaldi. "Effects of volume scaling in AlN nano plate resonators on quality factor." In *2016 IEEE International Frequency Control Symposium (IFCS)*, pp. 1-3. IEEE, 2016.

Nan, Tianxiang, Hwaider Lin, Yuan Gao, Alexei Matyushov, Guoliang Yu, Huaihao Chen, Neville Sun et al. "Acoustically actuated ultra-compact NEMS magnetoelectric antennas." *Nature communications* 8, no. 1 (2017): 1-8.

Regarding the impact of the ground ring on the device performance and its contribution in the energy harvesting experiment, please find below the explanation that is added to the paper:

“The ground ring shown in Fig. 2a forms two symmetrical loops with the center path, where the two ground pads are shorted by wire-bonding and the middle pad is the signal. As it was shown in the previous article by our group [15] the ground ring has a negligible contribution and impact on the ME antenna’s performance. This is mainly because the electromotive force generated by these two loops cancel each other and the net contribution is insignificant.”

The impact of the ground ring is widely investigated by our group in many different devices and it was also shown in [15], where two devices with similar size and ground rings were discussed, one of the devices with magnetic material and the other without magnetic material. It was shown that the device with magnetic material had a significantly higher performance compared to the device without magnetic film, suggesting that the ground rings do not have a big contribution to the output signal of the antenna.

Regarding the autonomous sensor system, we discussed this in the answer to the major comment 2 from reviewer 3 and added new contents in section 5 of the manuscript and appendix I, proposing an IC design for the magnetic sensing and data communication.

3) Fig. 3, PCB transmitter antenna is state of the art, the part is far too long for this journal.

Thanks for your comment. We have addressed this concern by removing the TX coil design part from the main manuscript. We have moved both the transmitter plots in Fig. 3 and the transmitter coil design explanation to the Appendix F in supplementary materials.

Part 2: Minor Comments and Questions

Furthermore, a number of questions related to the experiments presented should be answered before publication:

1) In section 2 the specification of a thickness of the resonator or the question of whether there is a remainder of the substrate under the layer stack is not addressed, but seems relevant to me.

Thank you for your comment. We have addressed these points in the Appendix B of supplementary materials where we discussed the micro-fabrication process. The silicon underneath the rectangular resonators is completely etched and there is no leftover. In

addition, the thickness of the AlN and FeGaB layers are both 500nm, giving a total thickness of 1um for the resonator.

2) Please correct the wording (line 125): no peaks are seen, three principal dips are visible in each plot of Fig. 2 b and c.

Thanks for your comment. We have fixed the wording.

3) Please provide information about the used network analyzer (line 130).

Thanks for your comment. We have provided the model of the network analyzer used in the experiments in the Appendix E of supplementary materials where we discussed the energy harvesting experimental setup. The network analyzer model we used is Agilent Technologies E8364A.

4) In sections 2-1 and 2-2 there is no schematic illustration of the measurement setup. Fig. 4 would have to refer to the receiving antenna (or the ME resonator) and be plotted as directional characteristic 3D - then it would be much easier to understand.

Thanks for your comment. The schematic illustration of the energy harvesting experimental setup is shown in Appendix E of supplementary materials. If the reviewer means we need to show the schematic diagram in the manuscript, we would be glad to make a schematic with a proper graphic and visualization and show it in the main manuscript.

Regarding the section 2-2, we have used the same experimental diagram shown in Appendix E, where we used two plastic manipulators that are adjustable along XYZ axis. The Tx coil and Rx antenna are mounted on these plastic manipulators and connected to the network analyzer (as an RF source) and power spectrum analyzer, respectively.

Regarding the Fig. 4, the reason we have included the Tx coil layout in the plots and used it as a reference is that the magnetic field and measured power transfer efficiency patterns are highly dependent on the Tx coil geometry. We have added the direction of the ME antenna in each plot to better relate the 2D field distribution from Tx coil to each angle of ME antenna. We believe that the directional characteristic 3D plots would eliminate the directionality and the impact of Tx coil geometry.

5) In section 2-3, quite unrealistic distances (all in the sub-cm range) are used and the modeling is simply not explained in more detail. Here a very complex field is broken down on two and a half pages.

Thanks for your comment. The sub-cm range is good enough for animal research, as shown in our manuscript by extracting the tissue layers of an adult rat and placing the Tx and Rx between them. Therefore if our understanding is correct, you are talking about unrealistic distances with regards to human beings, in which case you are correct. Although we can easily increase the Tx-Rx distance by simply increasing transmitted power, it is not the focus of our work at the moment. We realize this can be confusing as one long-term objective is to have a working prototype for deep brain implantation in human subjects, but a good starting point is a validation in animal work. Furthermore working with large distances would make comparisons with similar work that use sub-mm receivers more challenging (see Table 1).

Regarding the modeling, we have added Appendix G to the supplementary materials and have discussed more details on the modeling and added the electromagnetic properties of the tissue layers used in section 2-3.

6) The FOM (equation 1) that is introduced is very simplified, knowing the numerical values of the parameters is assumed, A remains unclear due to the unknown thickness of the resonator. More importantly, d is cubic and is treated here as a simple scalar, but is usually strongly anisotropic for any antenna with directional characteristics.

Thanks for your comment. We realize that a perfect FOM does not exist. We have chosen this FOM because it seems to be the preferred and most common method to evaluate coils operating in the near field regime. A is the top surface area of the ME antenna, we have made it more clear in the table 1. We have ignored the thickness ($\sim 1\mu\text{m}$) as its impact on the overall implant volume is negligible, and the same is true for the on-chip coils.

Although the ME is an antenna, for the sake of comparison we have treated it as a coil since all references included in Table 1 use an on-chip coil Rx.

7) Eq. (4) is set up, but no further reference is made, and the introduced x_{ME} never occurs again. The role of the bias field of exactly 37 Oe is not

discussed further, it is only mentioned once. Unusual to put it along the short axis.

Thanks for your comment. Equation 4 is derived from the third term (highlighted) of Equation 3, which is the strain due to the mutual piezoelectric/piezomagnetic effects or the strain-mediated ME effect x_{ME} .

$$x_{ij} = d_{ijk}^e E_k + d_{ijk}^m H_k + \pi_{ijkl} E_k H_l, \quad (3)$$

$$x_{ME} = \pi_{ijkl} E_k H_l$$

In this equation, $E = \frac{V}{t_{AIN}}$, where $V = V_c \cos(\omega_c t)$ and t_{AIN} are the applied RF voltage to and thickness of the AlN layer, respectively. And the external magnetic field can be expressed as $H = H_m \cos(\omega_m t)$. Therefore, we can simplify the third term in equation 3 as follows to demonstrate how the modulation effect occurs:

$$x_{ME} \propto \frac{1}{t_{AIN}} V_c \cos(\omega_c t) \times H_m \cos(\omega_m t) \quad (4)$$

$$\propto \frac{H_m V_c}{2 t_{AIN}} [\cos((\omega_c + \omega_m)t) + \cos((\omega_c - \omega_m)t)]$$

Therefore, effect x_{ME} , derived from the third term in equation 3 is simplified and mentioned in order to demonstrate how the external magnetic field is modulated on the applied RF signal. And that is why it is not occurs again.

Regarding the DC bias field of 37Oe, we have added below statement to the manuscript in order to make it more clear why exactly this field value is used.

“We have investigated the impact of the external DC bias field on the sensitivity of the sensor and applied DC fields from 0 to 70 Oe. The sensor performance and sensitivity were at their peak under the external DC bias field of 37 Oe.”

Regarding the direction of the DC bias field which is along the short axis, it is correct that in most other works on ME sensors—including previous works from our group—the DC bias field is along the length axis. We have applied DC bias field along both width and length axis and realized that the sensor’s performance is significantly better when the field is along the width. We believe this is because of the shape anisotropy and large length to width ratio of the resonators (5 to 1) in our work where the magnetic domains and easy-axis have oriented towards the length. In our previous works, with similar materials and fabrication procedure, the length to width ratio was in the range of

2~3 and that is why sensors showed better performance with DC bias field along the length axis. In those sensors, too, sometimes sensors perform nearly equally when the field is along the width or length axis. In this work, however, because of the large length to width ratio, sensor's performance was substantially better with the DC bias field applied along the width.

8) Fig. 6 is not so easy to obtain with the measuring device mentioned in the supplement, the plot (a) shows an incredibly large dynamic range of over 200 dB (corresponding to 10 orders of magnitude), the data sheet confesses a dynamic range of 100 dB (briefly above 6 orders of magnitude).

Furthermore, the noise falls considerably faster than $1/f$.

9) The noise carpet in Fig. 6 (a) and (b) is two orders of magnitude below the best case input noise voltage of $4 \text{ nV} / \sqrt{\text{Hz}}$ specified by the same measuring device.

11) I have also doubts about the given noise levels: Voltage densities in the $\text{pV}/\sqrt{\text{Hz}}$ -range seem to be very unrealistic to me. In a 50 ohm system, 30 or 40 $\text{pV}/\sqrt{\text{Hz}}$ corresponds to at a temperature of less than 1 K.

Thanks for the comments. The answer to the all three questions above (question 8, 9, 11) is explained as follows:

The reason for the large dynamic range in Fig. 6, the rapid noise attenuation, the noise voltages lower than $4 \text{ nV}/\sqrt{\text{Hz}}$, and the Voltage densities in the $\text{pV}/\sqrt{\text{Hz}}$ -range is attributed to the internal lock-in amplifier demodulator's filter—a low-pass-filter with a cutoff frequency of 89Hz—which is deployed to improve the SNR and LoD. *We have updated the manuscript and mentioned the use of low-pass filter at the input of lock-in amplifier.* Even though the low-pass filter theoretically attenuates both the signal and noise level with the same gain, we have seen 1~2 times improved SNR and LoD when the low-pass-filter is enabled. Even if we do not use the low-pass filtering feature the limit of the detection of the ME antenna should still be sufficient for the neural magnetic field recording according to the data published in [23] and [24].

We have also talked to the Zurich Instrument company about all the mentioned concerns in the questions above and they have confirmed that the results in the manuscript—including the noise values lower than $4 \text{ nV}/\sqrt{\text{Hz}}$ mentioned in the datasheets as the best case input noise and the $\text{pV}/\sqrt{\text{Hz}}$ range voltage densities—are valid and attainable when the low-pass filter option is enabled.

The low-pass filter suppresses the voltages above the cutoff frequency by a few orders of magnitude. In Fig. 6 we are demodulating an ~ 63 MHz signal with 1 kHz sidebands, but the 3dB bandwidth is only set to at 89 Hz. All signals outside of that including the noise and modulation peaks from the carrier will be suppressed.

10) If one calculates the sensitivity (i.e. slope) shown in Fig. 6 (c) one receives only about 10 mV/T (a standard hall sensor has about 10 V/T), please comment. Furthermore, the sensitivity of 10 mV/T and the noise density of 40 pV/sqrt(Hz) would result in a LOD of only 4 nT/sqrt(Hz).

Thank you for your comment. Regarding the sensitivity, the sensitivity (the slope) of the smart ME antenna is different at different ranges of applied magnetic field. The sensitivity increases by increasing the amplitude of the magnetic field. This is because the nonlinear effect gets stronger at higher magnetic fields. In Fig. 6 c, when applied magnetic field is in the range of nT, the sensitivity is about 10 mV/T. However, if we calculate the sensitivity based on the voltage peak value in Fig. 6b, which is under 9.4 uT magnetic field, we get a sensitivity of about 66 mV/T/sqrt(Hz). Moreover, it is not standard to compare the sensors solely based on the sensitivity of the sensors as this figure highly depends on multiple parameters such as frequency, field range, etc. In the case of ME antennas, for instance, the sensitivity at the resonance is significantly larger than the off-resonance sensitivity. For example, in the paper below from our group the on-resonance frequency of discussed ME antenna is about 38,000 V/T. In the sensor discussed in our paper, the derived sensitivity is the off-resonance figure and that is why it is low. It is notable that this ME antenna is big (cm-size), but the same principles are still valid for our μm -size antenna as well.

Dong, Cunzheng, Yifan He, Menghui Li, Cheng Tu, Zhaoqiang Chu, Xianfeng Liang, Huaihao Chen et al. "A Portable Very Low Frequency (VLF) Communication System Based on Acoustically Actuated Magnetolectric Antennas." IEEE Antennas and Wireless Propagation Letters 19, no. 3 (2020): 398-402.

Regarding the calculated LoD, as mentioned above, the sensitivity of the sensor changes at larger field values and we cannot calculate the LOD only based on the sensitivity at nT range fields (Fig. 6c). If we calculate the sensitivity over a larger magnetic field range (e.g. based on the data in Fig. 6b, where the dV is about 620 nV/sqrt(Hz) and dB is about 9.4 uT), we get a sensitivity of about 66 mV/T/sqrt(Hz). If we consider the noise density of about 40 pV/sqrt(Hz), the calculated LOD would be approximately 600 pT, which is comparable to the reported 470 pT figure at 1kHz.

Second explanation for LOD using SNR figure: based on the data in Fig. 6b, we can calculate an SNR of about 15,500 (peak value divided by the noise floor). The LOD of the sensor would be the point where the SNR is about 1. In other word, we can reduce the 9.4 uT field by 15,500 times before it is lost in the noise. The calculated LOD based on this explanation would be about 600 pT, which is comparable to the 470 pT figure reported in the paper.

In summary, because the sensitivity of the sensor changes depending on the applied magnetic field range, we cannot rely only on the sensitivity (slope) of the curve shown in Fig 6c to calculate the LOD.

12) The LOD in Fig. 6 (d) is shown to be flat between 250 Hz and 1500 Hz. The noise (taken from Fig. 6 (a)) increases about three orders from 1500 Hz to 250 Hz (from 40 pV (at 1500 Hz) to about 40 nV (at 250 Hz)), please comment.

Thanks for your comment. As discussed in the answers for comments 8, 9, 11, both the low-pass filter and 1/f noise reduction are the reason for noise attenuation. At frequencies below 250 Hz, electronics noises are more dominant according to our experiments and noise floor drops more rapidly than the ME modulation peak, which is why the LOD improves in this region. After the frequencies around 150~250 Hz, both the noise floor and ME modulation peak attenuate with almost the same rate, which is why the SNR or LoD become flat. We have made this more clear in the manuscript.

.....
.....

Reviewer 2 (Written in green color in revised manuscript and supplementary materials)

The work is devoted to the investigation of the ultra-compact dual-band Smart NEMS ME antennas for simultaneous wireless energy harvesting and magnetic field sensing. The Authors applied the original method and combined the systems of wireless energy harvesting and measuring of weak magnetic fields in one ME antenna. They used the operation of the ME antenna in two modes (thickness and width) at two frequencies of 2, 51 GHz to collect energy and 63.6 MHz for measuring the magnetic field in neural activities. They examined the issue of the disorientation of Tx and Px in the operating mode and presented experimental data on energy extraction in the study of the brain of mice. The ME antenna developed has shown high results; in size and sensitivity it

exceeds the known similar systems on micro- coils. But the Authors need to answer a number of questions.

The reviewer believes that after answering the questions posed, the article can be published in the Nature Communications

1) How justified is the choice of frequencies for the thickness and the width modes?

Thanks for your comment. The frequencies of ME antenna are defined by the width and the thickness of the resonators. In this set of fabricated ME antennas, we designed the devices mostly with a focus on different geometries and different length to width ratios of 4:1, 5:1, and 6:1. Usually a good length to width ratio for a resonator is in the range of 4 to 6. In this specific design the dimensions of each resonator are $250 \times 50 \times 1 \mu\text{m}^3$, providing a length to width ratio of 5:1.

For the thickness mode frequency of 2.51GHz, we wanted it to be close to the standard 2.45 GHz and 2.54 GHz wireless communication frequencies, but a bit different to reduce the interference effect. For the width frequency of 63MHz, we did not have a specific reason and it was mostly defined by the ratios that we chose, as well as keeping a large distance from the 2.51 GHz frequency.

However, both the width and thickness mode resonance frequencies can be readily changed by changing the dimensions of the resonators. In the future generation of devices, we aim to choose the frequencies in the standard industrial, scientific and medical (ISM) band, such as 40.68 MHz for width mode and 2.4~2.5 GHz for the thickness mode.

2) How to improve the signal spectrum in Figure 2?

Thanks for your comment. The plots in Fig. 2 show the S11 of the thickness and width mode of the ME antenna, where the dips show the resonant frequencies of the device. If you mean how to improve the S11 such that there is only one strong S11 dip for each mode and eliminate other smaller dips, we can achieve that by making sure that the stress level in the rectangular resonators is distributed uniformly, which could sometimes be challenging. The reason we have multiple dips in the S11, as mentioned in the manuscript, is that three rectangular resonators have slightly different stress levels which leads to different S11 dips.

If you mean how to improve the impedance matching and have deeper S11 dips, this can be achieved by optimizing the anchor width and length of the resonators, leading to a better impedance matching and improved S11 dips.

3) How is the area size Rx obtained in Table 1?

Thanks for your comment. The Rx area for ME antenna is the total area of the ME resonators which is $250 \times 174 \mu\text{m}^2$. We have made this more clear in the manuscript.

Reviewer 3 (Written in red color in revised manuscript and supplementary materials)

This manuscript by Sun and colleagues describes the latest developments in their NEMS magnetoelectric (ME) antennas for bioelectronic devices with important new capabilities. Specifically they describe a miniature device with wireless energy harvesting capability and impressive magnetic field sensing ability. The major interest in ME antennas is their ability to achieve good power densities even when the devices are made very small. This is an important property for implanted bioelectronics because small form factor devices are less invasive and enable potentially many independent devices in a small area of tissue. An exciting aspect of this paper is the fact that the same ME antenna used for power harvesting can also be used for magnetic field sensing. Magnetic field sensing is one important method to track activity of electrogenic cells like those in the brain.

Despite the important impact of the advancements discussed by the authors, the manuscript in its current form falls well short of the claims in a few important ways. These are summarized as major points below and followed by minor technical points that should be addressed in any potential revision.

Part 1: Major Comments

1) No simultaneous energy harvesting and sensing: Most importantly, the title suggests “Simultaneous energy harvesting and magnetic field sensing;” however, there appears to be no demonstration of simultaneous operation of power harvesting and magnetic field sensing. The details of

the magnetic sensing experiment are not clearly described, but it appears that for the sensing experiments the device is powered via direct electrical connection. The same antenna appears capable of sensing and power harvesting, but there may be difficulty in performing both functions simultaneously and certainly needs to be demonstrated if the authors are to use their current title.

I would suggest the authors significantly revise their claims and the title to better match what they actually show: the same antenna can function to sense magnetic fields and harvest data - although they do not show the ability to do so simultaneously. Else they should add experiments that clearly show simultaneous power and sensing.

Thanks for pointing this out. We have done the simultaneous energy harvesting and magnetic field sensing experiments as you described and have added another section at the end of the manuscript describing the results of the experiment. The diagram of experimental setup used in this experiment is also shown in Appendix H in the supplementary materials. The results are also shown in the plots in Fig. 1 below, where you can see the harvested power and magnetic field sensing performance at three different scenarios: 1) the device was operating only at its magnetic field sensing mode where the 2.51 GHz Tx coil was off and there was no splitter in the circuit; 2) the device was operating at its energy harvesting mode where there was no 63.6 MHz excitation, both the Helmholtz coils were off, and the coupler and splitter were disconnected; 3) the device was operating at both the energy harvesting and magnetic sensing mode where the 63.6 MHz excitation and the 2.51 GHz Tx coil as well as the DC and AC Helmholtz coils were all enabled. The coupler and the splitter were also added to the circuit, as shown in the Fig. S8 in Appendix H. For the energy harvesting tests we recorded the received power by the ME antenna at 2.51 GHz, and for the magnetic field sensing we recorded the ME modulation peak voltages, which are the most important parameter showing the performance of the sensor. The results from the simultaneous experiment showed that the received energy harvesting power at 2.51 GHz was 29% lower compared to the case where only energy harvesting mode was on and there was no splitter in the circuit. Moreover, the magnetic sensing data from the simultaneous experiment showed that the ME modulation voltages were 45% lower compared to the case where only magnetic field sensing mode was on and there was no splitter in the circuit. We realized that if we keep the splitter in the circuit and turn off the 2.51 GHz Tx coil we observe the same 45% reduction in the ME modulation voltage, and if we turn off the 63.6 MHz RF excitation as well as DC and AC test fields we still have 29% reduction in the harvested power. This observation implies that the act of enabling both the frequency modes of the antenna simultaneously does not degrade the performance

of the device—and that the observed signal reduction is solely associated to the presence of the splitter in the circuit which divides the harvested power and ME modulated signal into two lines.

It is notable that even after reduced PTE and magnetic field sensing performance in the simultaneous experiment the wireless energy harvesting FOM of the ME antenna is still higher than that of other works in the table 1. And the magnetic field LoD of the sensing mode is still enough to detect the magnetic neural amplitudes that are mentioned in the [23] and [24].

Figure 1. a) ME modulation voltage as a function of frequency at two different scenarios where ME antenna is operating at its sensing mode alone (blue) and where it's operating at both sensing and energy harvesting modes simultaneously (red); b) the zoom-in plot showing the amplitude of 1kHz modulated signal. The amplitude of ME modulated signal is reduced by 45% when the device is operating at both sensing and energy harvesting mode simultaneously; c) 2.51 GHz harvested power at two different scenarios where ME antenna is operating at its energy harvesting mode alone (blue) and where it's operating at both sensing and energy harvesting modes simultaneously (red). As it is shown the harvested power is reduced by 29% when the device is operating at both modes simultaneously. The reduced ME modulated voltage and harvested power is due to the added splitter in the simultaneous test which divides the signals into two lines, as it is shown in the schematic of experimental setup in Fig. S8 in Appendix H.

2) No uplink to communicate data to the Tx/Rx: As a sensor for an implantable bioelectronic device it is critical to communicate the sensed data back to the Rx. This is a critical element for 1) magnetic sensing, and 2) aligning the transmitter to device (as described on line 222-226).

The authors should clearly identify this need, and discuss potential solutions. Without this element the manuscript is misleading in the sense that much of the capabilities described could not be implemented in any practical way without a significant engineering advance.

The authors should clearly state the shortcomings of their current device and need for future work.

Thank you for bringing up that readers will benefit from further explanations regarding the envisioned integrated system under development. We have explored data transmission via a circuit design technique. The proposed method is to synthesize a local oscillator (LO) using the ME antenna operating at its contour mode resonator (CMR) resonance frequency, and then to use this LO to apply a sinusoidal-like excitation current at the ME antenna. This produces an amplitude-modulated output when there is a change in the magnetic field due to neuronal activity. As visualized in Fig. 2 below, the device with the ME antenna will require circuitry for energy harvesting at the FBAR resonance frequency, as well as a driver circuit that generates a carrier signal at the CMR resonance frequency for transmission of the sensed information. The magnetic modulation-based capability to transmit sensed data from the ME antenna with the described approach has been assessed by simulating an ME antenna model with the driver transistor stage (M_1), for which our results are published in the following paper:

[R1] I. Martos-Repath, A. Mittal, M. Zaeimbashi, D. Das, N. X. Sun, A. Shrivastava, and M. Onabajo, "Modeling of magnetoelectric antennas for circuit simulations in magnetic sensing applications," in *Proc. IEEE Intl. Midwest Symp. on Circuits and Systems (MWSCAS)*, Aug. 2020.

With the above-mentioned transmission of the sensed information, the required external receiver can perform standard operations for processing the amplitude-modulated signal generated in [R1]. The magnetically amplitude-modulated received signal has to be amplified and down-converted by the RF receiver front-end, and further processed with digital filtering to extract the signal components that carries the sensed magnetic field information. For the first implementation of these operations, we suggest the use of Universal Software Radio Peripheral (USRP) for software-defined radio processing of the amplitude-modulated signal.

We agree that the implementation of such an integrated device requires further engineering work that is outside of the scope of this paper—which focuses on "a compact, dual-band ME antenna for wireless energy harvesting and magnetic field sensing". While our groups have designed integrated circuits for energy harvesting and timing circuits, the

implementation of the envisioned system is now identified as future work in section 5 in the manuscript.

Fig. 2. Conceptual diagram of an ME antenna integrated with a chip for autonomous sensing and energy harvesting.

In addition, using the proposed IC design we can also align the external Tx coil to the implanted device in order to get the highest wireless power transfer efficiency. The device generates the largest power when the ME antenna is in the most optimum angle for the energy harvesting, which in turn it transmits back the strongest amplitude-modulated signal. In other word, based on the strength of the externally received amplitude-modulated signal we can align the Tx coil to the implanted device to achieve the highest power transfer efficiency.

3) Well short of in vivo practicality: All of the shortcomings would come through if the authors would perform an in vivo experiment. From what I can tell from the manuscript they are many missing pieces before that could happen. I do not necessarily think they need to complete these experiments before publishing, but the elements that need to be developed before implementation should be clearly stated (e.g. uplink, packaging, angular tolerance etc.). Without this clear and open discussion the manuscript is highly misleading.

Thank you for bringing up these points. We believe we have discussed these points in the previous comment and added new content on them to the manuscript as well, where we discussed our proposed design for the data communication, angular alignment, and packaging concerns.

Part 2: Minor Comments and Questions

1) Line182-184, Fig3-c: The device has a low output power - I think it's lower than the power required for stimulation / IC powering-, what applications could it be used for?

Thanks for your comment. By increasing the applied power to the external Tx coil we can receive larger powers at the ME antenna, in the range of few mW, which is enough to power the IC circuit which includes energy harvesting block as well as timing and control units, and to provide enough charge for neuromodulation. For instance, at 5mm distance between ME antenna and Tx coil, the wireless PTE is approximately 0.1%. If we apply a 30dBm signal (1W) to the Tx coil we can receive 1mW at the ME antenna. This received power is larger than the power many state of the art miniaturized wireless devices operate at for IC powering and neural stimulation, for instance the devices in [11], [25], [27] which operate with 100s of micro-watts power. It is noteworthy that the focus of this manuscript is on magnetic sensing, but in the future works we aim to investigate the neural stimulation feature as well.

2) Fig6, Fig S6, line 348-352: For the testing setup of the magnetic field sensing, a directional coupler was used to carry the reflected signal from the ME antenna that contains the strain-mediated ME response, how the setup will change for the implanted device?. Also, the ME-Modulated voltage (Fig6-c) was extremely small, -if not carried by the coupler- how to ensure robust operation of the film?

Thanks for your comment. Please note the approach described in the previous comments where we discussed our proposed circuit design for data communication.

Regarding the small amplitude of the ME modulated voltage, the results shown in the manuscript are after the lock-in amplifier post-processing and low-pass filtration, which reduce the amplitudes. Pre-filtered ME-modulated voltages are in the range of microvolts. This point is discussed in the answers to the comments 8, 9, 10 from the first reviewer and added to the manuscript as well. The internal low-pass filtration by the lock-in improves the SNR by a factor of 1~2. Even if we do not use the low-pass filtering feature the limit of the detection of the ME antenna should still be sufficient for the neural magnetic field recording according to the data published in [23] and [24].

3) Line 110-114: The formula used to calculate the resonance frequency -for both the width and thickness modes- depends only on the properties of the piezoelectric material, however, in other literature it is a function of both the piezoelectric and the magnetostrictive properties.

Thank you for pointing this out. We made a mistake in the manuscript in that sentence. We have in fact used the weighed equivalent density and equivalent young's modules of the resonator stack, and not the AlN layer alone. This is the standard method that is widely used in the MEMS community.

4) Fig6-a: What does the amplitude of the carrier component represent?

Thanks for your comment. As you know, the amplitude of the career signal is the reflected component of the 63.6 MHz signal that we apply to the sensor to enable its sensing mode. The changes in the amplitude of the career signal (i.e. signals at frequency 0 Hz after demodulation) can be used to sense the external DC magnetic field. In the same way that the modulation method works, where the changes in the 1 kHz external magnetic field changes the amplitude of 1 kHz signal on the spectrum, the 0 Hz signal also corresponds to the amplitude of the DC magnetic field. Our group has in fact published two papers for DC magnetic field detection where we only investigated the amplitude of the career signal (or the post-modulation signal at 0 Hz). These articles are shown below:

Li, Menghui, Alexei Matyushov, Cunzheng Dong, Huaihao Chen, Hwaider Lin, Tianxiang Nan, Zhenyun Qian, Matteo Rinaldi, Yuanhua Lin, and Nian X. Sun. "Ultra-sensitive NEMS magnetoelectric sensor for picotesla DC magnetic field detection." *Applied Physics Letters* 110, no. 14 (2017): 143510.

Nan, Tianxiang, Yu Hui, Matteo Rinaldi, and Nian X. Sun. "Self-biased 215MHz magnetoelectric NEMS resonator for ultra-sensitive DC magnetic field detection." *Scientific reports* 3, no. 1 (2013): 1-6.

Reviewers' Comments:

Reviewer #1:

Remarks to the Author:

The authors have addressed my comments 1-7 sufficiently.

Unfortunately the quite important points 8, 9, 11 were addressed rather casually.

It is a rather unlogical procedure to apply a low-pass filter with a corner frequency of 89 Hz around a signal at 63 MHz, when one is actually interested in demodulating a signal at 1 kHz on those 63 MHz. This leads to tremendous, artificial damping of the signal as well as of the noise at 1 kHz off-carrier. Plotting this situation without correction for the filter effects is strongly misleading, as the noise floor reaches unphysical values on the order of $\text{pV}/\sqrt{\text{Hz}}$. Thus it is of prime interest to the reader to know the exact filter type and its order, as these will add to the bandpass filter characteristic of the mechanical resonator, making distinction between both very difficult.

The comments 10 and 12 arose because of the lack of information disclosed on the sensor signal flow, see 8, 9 and 11.

Taking into account the effects of a misplaced low-pass filter in the instrumentation this comment may seem invalid, nevertheless a comparison to data provided by literature following common standard procedures in terms of signal processing remains impossible in a straight forward manner.

Reviewer #2:

Remarks to the Author:

The authors gave exhaustive answers to my questions and I am satisfied with their answers. In general, the revised and supplemented article is well read and clearly shows the broad prospects for the application of magnetoelectric composites in biomedicine. I believe that an article in this form can be published in the journal "Nature Communications".

Reviewer #3:

None

Point-by-Point Response Letter for Paper ID: NCOMMS-20-15513A

Dear Reviewers and Editors,

We would like to appreciate the time and efforts you have put to provide the perceptive feedback to improve the quality of our paper.

We are pleased that we were able to satisfactorily address most of the comments in the previous submission, and that we have this opportunity to provide the answers to the 2nd reviewer's follow-up comments. Please note the point-by-point responses below, for which the corresponding changes in the manuscript are emphasized with red text color.

Reviewer #1 (Remarks to the Author):

Unfortunately the quite important points 8, 9, 11 were addressed rather casually. It is a rather unlogical procedure to apply a low-pass filter with a corner frequency of 89 Hz around a signal at 63 MHz, when one is actually interested in demodulating a signal at 1 kHz on those 63 MHz. This leads to tremendous, artificial damping of the signal as well as of the noise at 1 kHz off-carrier. Plotting this situation without correction for the filter effects is strongly misleading, as the noise floor reaches unphysical values on the order of $\text{pV}/\sqrt{\text{Hz}}$. Thus it is of prime interest to the reader to know the exact filter type and its order, as these will add to the bandpass filter characteristic of the mechanical resonator, making distinction between both very difficult. The comments 10 and 12 arose because of the lack of information disclosed on the sensor signal flow, see 8, 9 and 11. Taking into account the effects of a misplaced low-pass filter in the instrumentation this comment may seem invalid, nevertheless a comparison to data provided by literature following common standard procedures in terms of signal processing remains impossible in a straight forward manner.

Reply: Thank you for bringing up that further clarifications in the manuscript will aid the readers. With the 89Hz cutoff frequency, the main signal, noise and the modulated signal are all reduced by approximately 66dB, which is now stated in Section 4 of the manuscript. We performed evaluation measurements by applying a signal modulated with a 1kHz sinusoid to the lock-in amplifier in order to compare the 89Hz and 1.5kHz low-pass filter frequency settings. The 89Hz cutoff frequency setting was used because we observed a slightly higher SNR when integrating the noise around the signal of interest. Please note the following additional information that explains the low noise floor:

- The lock-in amplifier has a digital low-pass filter option with adjustable bandwidth and order (from 1 to 8). This filter is applied on the digitized signal after digital mixing of the signal and reference frequency. We have measured the attenuation of the filter at 1kHz with two different cutoff frequency settings, and compared the results with the calculated expectations based on the frequency response using equations provided by Zurich Instrument [1]:
 - The theoretical filter response at 1kHz when the 3-dB bandwidth of filter is 89.13 Hz (corresponding to $\tau = 776.7\mu\text{s}$ and order = 4) is equal to 0.0016
 - The theoretical filter response at 1kHz when the 3-dB bandwidth of filter is 1.502 kHz (corresponding to $\tau = 46.11\mu\text{s}$ and order = 4) is equal to 0.8511

→ The ratio of the response is: $0.8511/0.0016 = 524.1$

Fig. R1 and Fig. R2 below show the measurement results displayed with LabOne (i.e., the lock-in amplifier software interface) using the peak measurement tool:

- Signal at 1 kHz with 89.13Hz filter = $13.2\text{e-}9$ V
- Signal at 1 kHz with 1.502kHz filter = $6.93\text{e-}6$ V

➔ The ratio of the two measured cases is: $6.93e-6 \text{ V} / 13.2e-9 \text{ V} = 525$

It can be seen that the above theoretical calculations agree well with the measurements.

The input-referred noise of the lock-in amplifier is impacted by the analog front-end (pre-amplification) and the analog-to-digital conversion (ADC). This determines the ultimate sensitivity of the measurement. The displayed noise floor level is subject to the settings of the discrete-time RC filter in the lock-in amplifier. This discrete-time RC filter is essentially a moving average filter that contributes to a significant noise floor reduction. Fig. R1 and Fig. R2 reveal that the noise floor is reduced by 66dB. The filter information and noise floor reduction are now described in Section 4.

Reference:

[1] Description of Zurich Instruments' lock-in amplifier filter:

<https://blogs.zhinst.com/mehdia/2019/03/19/frequency-domain-response-of-lock-in-filters>

Fig. R1. Measurement under 89Hz low pass filter

Fig. R2. Measurement under 1.502kHz low pass filter

Reviewers' Comments:

Reviewer #1:

Remarks to the Author:

The authors have addressed my comments sufficiently.